# SARM: Stage-Aware Reward Modeling for Long Horizon Robot Manipulation

**Qianzhong Chen**[1,3,*,†], **Justin Yu**[2,3,†], **Mac Schwager**[1], **Pieter Abbeel**[2], **Yide Shentu**[2,3], **Philipp Wu**[3]
[1]Stanford University, [2]UC Berkeley, [3]xdof.ai
`{qchen23, schwager}@stanford.edu`
`{yujustin, pabbeel, fredshentu}@berkeley.edu`
`{davidchen, justinyu, philipp}@xdof.ai`

## Abstract

Large-scale robot learning has made progress on complex manipulation tasks, yet long-horizon, contact-rich problems—especially those involving deformable objects—remain challenging due to inconsistent demonstration quality. We propose a stage-aware, video-based reward modeling framework that jointly predicts task stage and fine-grained progress, using natural-language subtask annotations to derive consistent labels across variable-length demonstrations. This avoids the brittleness of frame-index-based labeling and provides stable supervision even in tasks like T-shirt folding. Our reward model is robust to demonstration variability, generalizes to out-of-distribution scenarios, and improves downstream policy training. Building on it, we introduce *Reward-Aligned Behavior Cloning (RA-BC)*, which filters and reweights demonstrations based on reward estimates. Experiments show that our method significantly outperforms baselines in both real-world rollouts and human validation. On T-shirt folding, we achieve 83% success from the flattened state and 67% from the crumpled state, compared to 8% and 0% with vanilla BC. Overall, our results highlight reward modeling as a scalable and annotation-efficient solution for long-horizon robotic manipulation. Project website: `https://qianzhong-chen.github.io/sarm.github.io/`.
*Keywords: Imitation Learning, Reward Modeling, Robotics Manipulation*

## 1 Introduction

The long-standing vision of enabling robots to seamlessly assist humans in household chores has inspired decades of research in robotics. From tidying living spaces to preparing meals, such capabilities hold the promise of freeing up human time, and improving quality of life. Recent progress in foundation models for robotics, or more generally robot behavior models (RBMs), has sparked renewed optimism toward this goal. By combining visual perception, motor control, and optionally language processing in a single framework, RBMs (Chi et al., 2023; Zhao et al., 2023; Chen et al., 2025; Sun et al., 2024; Huang et al., 2024; Yu et al., 2024; Wang et al., 2023a; Black et al.; Team et al., 2024; Zitkovich et al., 2023; Shentu et al., 2024; Huang et al., 2025a;b) enable robots to perform complex tasks, making it possible to execute these tasks in unstructured household environments.

Despite their promise, RBMs still struggle with long-horizon, contact-rich manipulation, particularly with deformable objects like T-shirts. Such tasks demand handling changing geometries, occlusions, fabric variations, and error-free multi-step planning—challenges where current models, often tuned for short-horizon rigid-object tasks, fall short. They fail to generalize beyond curated data, lose consistency over time, and misinterpret intermediate states. While many prior works in RBMs have focused on scaling up data (Barreiros et al., 2025; Lin et al., 2024), far less attention has been given to data quality. However, high-quality data is difficult to obtain: expert demonstrations are costly and time-intensive, while larger datasets often include noisy or suboptimal trajectories from less experienced operators. Even more challenging, demonstration quality itself is a difficult

---

*Corresponding author.
†Work done during internships at xdof.ai.

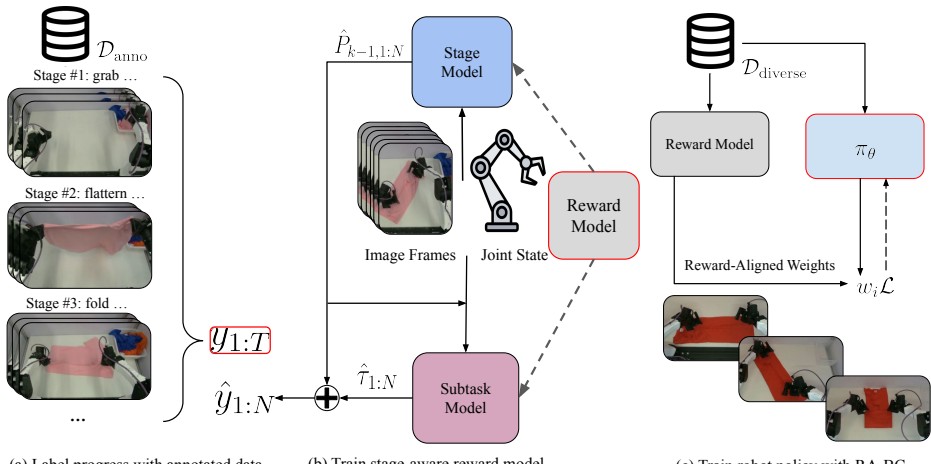

Figure 1: Overview of our method's framework for (a) data processing, (b) reward model training, and (c) policy training with reward signals. $\mathcal{D}_{\text{anno}}$ denotes the annotated dataset used for training the reward model, with examples shown in Fig. 5 and Fig. 6. $\mathcal{D}_{\text{diverse}}$ refers to a diverse expert dataset without annotations, which contains many suboptimal trajectories.

metric to quantify, since it depends on hidden factors such as action consistency and contact stability that cannot be directly measured, aside from simple proxy heuristics like task duration. Although there exist more sophisticated data-modeling approaches for assessing data quality and filtering trajectories after policy training (Belkhale et al., 2023; Dass et al., 2025; Agia et al., 2025), practical evaluation of demonstration quality remains challenging.

In light of these challenges, we propose a video-based reward modeling framework that leverages natural language annotations to assign progress labels and enable stable reward estimation for multi-step tasks. The learned reward model drives a Reward-Aligned Behavior Cloning (RA-BC) framework, filtering higher-quality data and improving policy performance in both simulation and the real world. Focusing on the T-shirt folding task, our experiments show that coupling the reward model with RA-BC significantly boosts performance, underscoring the importance of data quality in long-horizon manipulation. Together, these contributions advance scalable and annotation-efficient imitation learning. An overview is shown in Fig. 1.

Our contributions can be summarized as follows:

- We present SARM: a stage-aware reward modeling framework that automatically derives task progress labels from natural language annotations. Given any subsequence of RGB frames, the model jointly predicts the current task stage and fine-grained progress within that subtask, achieving robustness, generalization to out-of-distribution scenarios, and strong utility for downstream policy learning.

- We propose the *RA-BC* framework, which leverages the learned reward model to identify high-quality demonstrations and reweight training data accordingly.

- We validate our approach on the real-world task of T-shirt folding, a challenging long-horizon task that requires manipulating deformable objects, where it consistently outperforms strong behavior cloning baselines.

## 2 RELATED WORKS

### 2.1 LEARNED REWARD MODELS FOR ROBOTICS

Prior work on learning reward functions includes inverse reinforcement learning (Ng et al., 2000; Abbeel & Ng, 2004; Ramachandran & Amir, 2007; Ziebart et al., 2008; Finn et al., 2016), which infers rewards from demonstrations but suffers from reward identifiability and sensitivity to partial observability that hinder scalability to high-dimensional, long-horizon problems.

Learning from human feedback (e.g., preference rankings, scaled preferences, interventions) has proven effective in training large language models (LLM) (Christiano et al., 2017; Ziegler et al., 2019). Recently, RLHF has also gained increasing interest in robotics but still requires substantial task-specific input and suffers from annotator inconsistency (Sadigh et al., 2017; Liu et al., 2023).

A complementary direction uses LLM to synthesize reward functions or shaping code (Ma et al., 2024a; Shentu et al., 2024), which can accelerate bootstrapping but often assumes privileged or structured state information that is rarely available outside simulation and can degrade under sensor noise and domain shift.

Several prior works (Lee et al., 2021; Ma et al., 2022; Escontrela et al., 2023) estimate rewards by computing the feature distance to a goal state, enabling self-supervised reward model training without manual annotation. While effective for simple tasks with a single objective, such approaches struggle in long-horizon settings where the task naturally decomposes into multiple subtasks or stages. In these cases, a single goal distance fails to capture intermediate progress, often causing the reward signal to become uninformative or misleading.

Another line computes rewards directly from visual observations combined with task text using vision-language models (VLM). Among these, LIV (Ma et al., 2023), VLC (Alakuijala et al., 2024), GVL (Ma et al., 2024b), VICtoR (Hung et al., 2024), REDS (Kim et al., 2025), ReWiND (Zhang et al., 2025) and SARM—reward robot manipulation tasks directly from visual perceptions. In practice, many VLM based reward models struggle on long-horizon, highly dynamic, and contact-rich manipulation tasks because they process entire trajectories from the initial frame to resolve temporal dependencies, which increases data and computation demands and impedes scaling.

There are prior works such as DrS (Mu et al., 2024) and REDS (Kim et al., 2025) that use stage-aware reward models for long-horizon tasks. However, DrS is fundamentally different from visual-based SARM: it is purely state-based, depends on full simulator states as stage indicators, and requires training a separate discriminator for each stage, making it difficult to scale. REDS also differs from SARM: instead of modeling a continuous frame-wise progress curve, it learns a semi-sparse step-shaped reward with monotonicity regularization, which struggles to generalize when trajectories progress at different speeds. In addition, REDS infers stage via image–subtask embedding similarity rather than a dedicated stage-estimation network, which becomes unreliable when subtask descriptions are semantically similar.

## 2.2 IMITATION LEARNING WITH SUBOPTIMAL DEMONSTRATIONS

Prior works have explored imitation learning under suboptimal datasets. One direction adopts bootstrapped frameworks (Sasaki & Yamashina, 2020; Belkhale et al., 2023; Dass et al., 2025; Agia et al., 2025), which actively change the dataset distribution based on analyzing current policy's gradient, rollout, or learning objective during training. While effective, such methods are computationally expensive and require extensive hyperparameter tuning. Another line of research focuses on explicitly labeling and classifying demonstrations (Wu et al., 2019; Wang et al., 2023b), but this approach depends on a small, high-quality dataset as prior knowledge.

An alternative direction investigates weighted BC through offline reinforcement learning (RL) techniques (Wang et al., 2018; Chen et al., 2020; Siegel et al., 2020; Xu et al., 2022), where estimates of the advantage function are used to prioritize actions in the dataset. These methods, however, often assume access to full-state feedback and a well-trained critic, and have not been validated on real-world, vision-based, long-horizon manipulation tasks. In contrast, our RA-BC framework leverages a pre-trained, vision-based reward model to generate robust and accurate $K$-step advantage estimates, which then guide weighted BC training.

## 3 METHOD

### 3.1 REWARD MODEL TRAINING

**Data Processing.** Extracting dense reward labels remains a challenge, especially in long-horizon, complex tasks. Prior work often relies on frame indices as labels (Zhang et al., 2025). While this may suffice for short tasks with fixed duration, such as "pick up the cup," it fails for tasks like

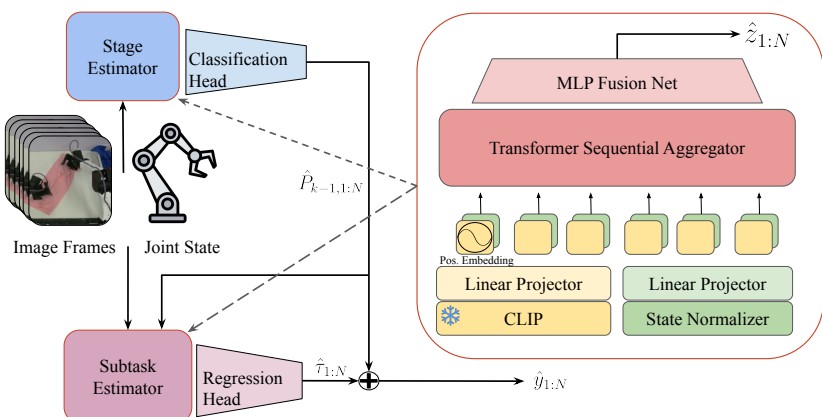

Figure 2: Overview of **SARM,** stage-aware reward modeling. **Left:** SARM overview, which includes both a stage estimator and subtask estimator. First the task stage is predicted from the observations. This prediction is additionally passed into the subtask estimator which predicts a scale value of the progress within the stage. **Right:** An overview of the estimator architecture which is replicated for both the stage estimator and the subtask estimator.

"fold the T-shirt," where trajectories vary greatly, task duration is not fixed, and motion sequences differ across demonstrations. For example, in T-shirt folding, the flattening phase may require more or fewer motions depending on shirt placement or fabric configuration, yet frame-based labeling only reflects elapsed time. As a result, identical task states (e.g., a fully flattened shirt) can receive progress values ranging from 0.2 to 0.8, introducing severe label noise that harms reward model learning and downstream policy training.

To resolve this, we leverage subtask annotations on the robot trajectory data. The collected trajectories consist of three video streams (top, left wrist, and right wrist), joint states, and joint actions. Before annotation, we designed annotation protocols by decomposing each task into semantically meaningful subtasks. For T-shirt folding, we developed two distinct protocols: one for sparse annotation and another for dense annotation, as illustrated in Table A.3 and Fig. 5 and 6. During annotation, only the subtasks defined by the protocol were labeled, and any trajectory that did not contain the complete sequence of subtasks specified by the protocol was discarded. Annotators watched the top-view video and segmented each trajectory into subtasks by recording the start and end frame indices. If a serious mistake (e.g. the manipulator hitting the table heavily or executing a completely reversed motion sequence) occurred during execution, its start and end frames were also labeled; trajectories containing mistakes were excluded from subsequent model training.

Using the annotated data, we computed the average temporal proportion of each subtask across the dataset to automatically assign progress values to the start and end frames of each subtask. Within each subtask, finer-grained progress labels were generated by linearly interpolating over frame indices. This procedure ensures that progress labels remain closely aligned with the semantic meaning of the motions while maintaining consistency across the entire dataset.

*Labeling by subtask priors:* Let a trajectory $i$ have total length $T_i$ and be segmented into $K$ subtasks with lengths $\{L_{i,k}\}_{k=1}^{K}$. We estimate a dataset-level prior proportion for each subtask

$$\bar{\alpha}_k = \frac{1}{M} \sum_{i=1}^{M} \frac{L_{i,k}}{T_i}, \qquad \bar{\alpha}_k \geq 0, \ \sum_{k=1}^{K} \bar{\alpha}_k = 1, \qquad (1)$$

where $M$ is the number of trajectories.

*Frame-wise progress targets:* For a frame $t$ that lies inside subtask $k$ with local bounds $[s_k, e_k]$, define the within-subtask normalized time $\tau_t = \frac{t-s_k}{e_k-s_k} \in [0, 1]$ and the cumulative prior $P_k = \sum_{j=1}^{k} \bar{\alpha}_j$ (with $P_0 = 0$). We assign the normalized progress target

$$y_t = P_{k-1} + \bar{\alpha}_k \tau_t \in [0, 1], \qquad (2)$$

so that $y_{s_k} = P_{k-1}$ and $y_{e_k} = P_k$.

**Model Architecture.** We adopt a dual reward-model architecture with a shared backbone architecture and two task-specific heads. The **stage model** predicts the current high-level stage, while the **subtask model** estimates fine-grained progress conditioned on the stage prediction. An overview of SARM architecture is demonstrated in Fig. 2. These models operate sequentially: the subtask model uses the predicted stage as prior context to refine the final progress estimate. The stage model outputs a probability distribution over discrete task stages, providing a coarse localization of the robot's progress, while the subtask model leverages the stage embedding to produce a continuous progress value in $[0, 1]$. Together, they provide both high-level stage classification and fine-grained progress estimation, enabling stable reward modeling in long-horizon manipulation tasks. An overview of the reward-model architecture is shown in Fig. 2.

The input pipeline proceeds as follows: (1) a sequence of $N$ images is encoded by a frozen CLIP encoder, producing visual embeddings shared across both models; (2) visual embeddings and joint states are projected into a common $d_{\text{model}}$-dimensional space, where only the first frame receives an explicit positional bias to prevent absolute temporal leakage, following ReWiND (Zhang et al., 2025); (3) the multimodal sequence is then processed by a transformer encoder to capture temporal dependencies and cross-modal interactions; (4) a lightweight MLP head fuses the aggregated features and outputs either stage logits $\hat{\Psi}_{1:N} \in \mathbb{R}^{N \times k}$ (stage model) or scalar progress predictions $\hat{\tau}_{1:N} \in [0, 1]^N$ (subtask model), where the latter is explicitly conditioned on the predicted stage to refine the progress estimate. Stage probabilities are obtained via a softmax $\Pi_{1:N} = \text{softmax}(\hat{\Psi}_{1:N}) \in [0, 1]^{N \times k}$, from which the discrete stage prediction and normalized progress are calculated as

$$\hat{S}_{1:N} = \arg \max_{i \in \{1, \ldots, k\}} \Pi_{1:N,i}, \quad \hat{S}_t \in \{1, \ldots, k\}, \tag{3}$$

$$\hat{y}_{1:N} = \hat{P}_{k-1, 1:N} + \bar{\alpha}_{k, 1:N} \hat{\tau}_{1:N}, \quad \hat{y}_{1:N} \in [0, 1]. \tag{4}$$

## 3.2 REWARD-ALIGNED BEHAVIOR CLONING (RA-BC)

Behavior Cloning (BC) trains a policy $\pi_\theta$ to imitate actions from demonstrations by minimizing a supervised loss on state–action pairs $(o_i, a_i)$. The standard BC objective averages per-sample losses,

$$\mathcal{L}_{\text{BC}}(\theta) = \frac{1}{N} \sum_{i=1}^N \ell(\pi_\theta(o_i), a_i), \tag{5}$$

where $\ell$ is mean squared error for continuous actions or cross-entropy for discrete actions.

**RA-BC objective.** RA-BC replaces the uniform prior in equation 5 with a *reward-aligned* weighting that emphasizes demonstrations predicted to make progress. For each training item $i$, we sample a *current* window (anchor) and its *next* window obtained by advancing one action chunk. Let $\phi(\cdot) \in [0, 1]$ denote the normalized progress score produced by the reward model (Sec. 3.1). If the anchor window ends at time $t$ and the chunk length (stride) is $\Delta$, we form a per-item progress *delta*

$$\hat{r}_i = \phi(o_i^{t+\Delta}) - \phi(o_i^t), \tag{6}$$

which serves as a scalar signal of expected improvement. This $\hat{r}_i$ is then mapped to a weight $w_i \in [0, 1]$ (see weighting rules below), and RA-BC minimizes the normalized weighted objective

$$\mathcal{L}_{\text{RA-BC}}(\theta) = \frac{\sum_{i=1}^N w_i \, \ell(\pi_\theta(o_i), a_i)}{\sum_{i=1}^N w_i + \varepsilon}, \tag{7}$$

with a small $\varepsilon > 0$ to avoid division by zero.

**Weighting from running statistics.** To calibrate $w_i$ without fixed heuristics, RA-BC maintains online running statistics (mean $\mu$ and standard deviation $\sigma$) of the raw progress deltas $\{\hat{r}_j\}$ via a numerically stable estimator (Welford). We clamp the running mean to be nonnegative, $\mu \leftarrow \max(\mu, 0)$, to avoid centering weights around negative progress in early training. Each $\hat{r}_i$ is mapped to a soft weight by a linear ramp between $(\mu - 2\sigma)$ and $(\mu + 2\sigma)$:

$$\tilde{w}_i = \text{clip}\left(\frac{\hat{r}_i - (\mu - 2\sigma)}{4\sigma + \epsilon}, 0, 1\right), \tag{8}$$

where $\text{clip}(x, 0, 1) = \min(\max(x, 0), 1)$ and $\epsilon > 0$ guards small variances.

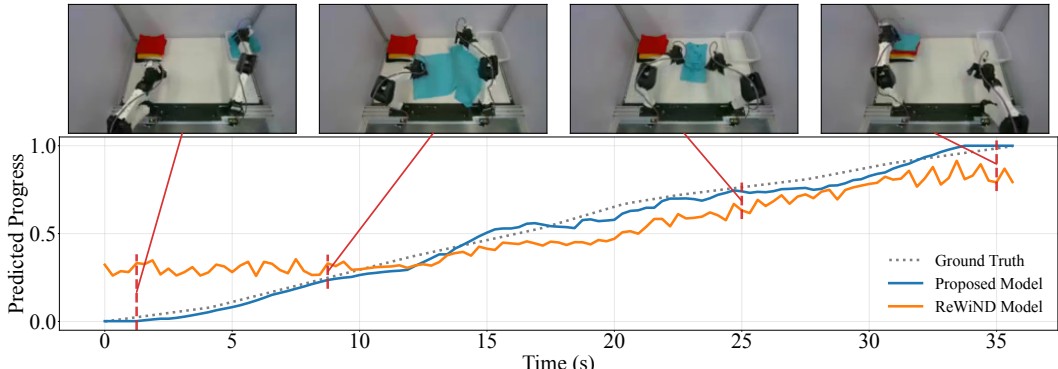

Figure 3: A visualization of the predicted task progress for T-shirt folding demonstrations. Compared with ReWiND, SARM provides more accurate and calibrated estimates.

**Prior overrides and validity mask.** We incorporate lightweight prior knowledge via a threshold $\kappa > 0$ to make weights decisive for clearly good/bad items:

$$w_i = \mathbf{1}_{\{\hat{r}_i > \kappa\}} + \mathbf{1}_{\{0 \le \hat{r}_i \le \kappa\}} \tilde{w}_i. \tag{9}$$

**Granularity and implementation.** RA-BC is architecture-agnostic and can be applied at the sample or sequence level. In our implementation, losses are first averaged over a temporal chunk to obtain a per-item loss, after which Eqs. equation 8–equation 9 produce $w_i$ used in Eq. equation 7. This makes RA-BC a drop-in replacement for Eq. equation 5 that *softly filters* noisy or non-progressing data while preserving training stability via normalization. In practice, RA-BC selectively emphasizes high-quality segments and down-weights suboptimal ones, enhancing policy learning especially when the dataset is diverse and contains imperfect demonstrations.

## 4 RESULTS

In this section, we answer three questions:

- **Q1.** How does SARM lead to more robust reward model when faced with long horizon, complex manipulation tasks?
- **Q2.** How can RA-BC enhance policy training when faced with diverse datasets?
- **Q3.** How does the quality of the reward model affect RA-BC performance?

### 4.1 Q1: REWARD MODEL EVALUATION

We evaluate SARM on two tasks, (1) T-shirt folding: a long-horizon, multi-stage, contact-rich manipulation problem, (2) unload dishes from a rack: a shorter-horizon multi-stage task with high variation due to varying dish counts, orientations, rack positions, and optional handovers. We utilized these two tasks to demonstrate the effectiveness of the proposed reward model training framework.

**Baselines.** We compare **SARM** against **LIV** (Ma et al., 2023), a reward model pre-trained on EpicKitchens (Damen et al., 2022); **VLC** (Alakuijala et al., 2024), which fine-tunes a VLM via sequential ranking to encourage monotonically increasing rewards; **GVL** (Ma et al., 2024b), which prompts a pre-trained VLM with shuffled frames to predict per-frame progress; VICtoR (Hung et al., 2024), which determines the motion class and motion progress by evaluating the similarity between the vision and text embeddings encoded from current frame and language instruction, respectively; REDS (Kim et al., 2025), which is a stage-aware reward model learning from the stage segmentation; and **ReWiND** (Zhang et al., 2025), which augments input sequences with rewound frames to improve robustness against failure cases. All applicable baselines use transformer encoders matched in model size to SARM; implementation details are provided in A.4. All baselines are trained on the union of the dense and sparse datasets. Ground-truth rewards are normalized to the range $[0, 1]$ for both annotation types.

Table 1: Evaluation of reward models. "Demo $\mathcal{L}$" denotes the single-step MSE of reward models on the validation set. All models are evaluated on 70 trajectories (50 from $\mathcal{D}_{\text{sparse}}$ and 20 from $\mathcal{D}_{\text{dense}}$), where both ground-truth progress and model predictions are normalized to the $[0, 1]$ range. The two-scheme models (last two columns) are evaluated in "sparse mode." "Rollout $\rho$" reports performance on real policy rollouts. Visualization examples of reward model predictions on both demonstration data and policy rollouts are provided in Appendix A.5.

| Metrics | Baseline Methods | | | | | | Ablation Studies | | | | SARM |
|---|---|---|---|---|---|---|---|---|---|---|---|
| | GVL | VLC | LIV | REDS | VICtoR | ReWiND | Dense | Sparse | w/o R | SARM (VB) | |
| Demo $\mathcal{L}$ ↓ | 0.064 | 0.083 | 0.021 | 0.036 | 0.079 | 0.019 | 0.027 | 0.013 | **0.008** | 0.015 | **0.009** |
| Rollout $\rho$ ↑ | -0.39 | -0.33 | 0.33 | 0.16 | 0.00 | 0.50 | 0.11 | 0.78 | 0.67 | 0.78 | **0.94** |
| | *Classification SR Breakdown* | | | | | | | | | | |
| SE | 0/12 | **12/12** | 6/12 | **12/12** | 0/12 | **12/12** | **12/12** | 10/12 | **12/12** | **12/12** | **12/12** |
| PSE | 6/12 | 0/12 | **12/12** | 8/12 | 3/12 | 8/12 | 3/12 | 11/12 | 9/12 | 10/12 | 11/12 |
| FE | 5/12 | 0/12 | 6/12 | 7/12 | **12/12** | 7/12 | 5/12 | 11/12 | 9/12 | 10/12 | **12/12** |

**Ablation Studies.** We additionally perform ablations by (1) **Dense**: training a single-scheme model only on the dense annotation dataset, (2) **Sparse**: training a single-scheme model only on the sparse annotation dataset, (3) **w/o R**: removing rewinding frames augmentation, and (4) **SARM (VB)**: evaluating SARM under varied brightness, where each frame's brightness is perturbed by up to ±0.3 from its original value.

**Evaluation Protocol.** Evaluation consists of two parts. First, for **human demonstration progress estimation**, models are evaluated on unseen testing data. For all baselines and two-scheme models, the validation set is the union of the 10% hold-out data from both datasets. For the single-scheme ablations, we apply cross-dataset validation: models trained on dense annotations are evaluated on the sparse hold-out set, and vice versa. We report single-step mean squared error (MSE) loss $\mathcal{L}$ on the validation set. Second, for **robot rollout progress estimation**, we fine-tune a **Pi0** policy (Black et al.) on the datasets we mentioned above using RA-BC, and then deploy the policy at different training stages on a real robot to collect 36 trajectories. These trajectories include 12 successful episodes (SE), 12 partially successful episodes (PSE), and 12 failed episodes (FE) rollouts. Reward models are evaluated on these rollout trajectories according to the following classification protocol:

$$\text{Label} = \begin{cases} \text{SE}, & \text{if } P_{\text{final}} > 0.8 \ \wedge \ \frac{1}{T/3}\sum_{t=2T/3}^{T} P_t > 0.6, \\ \text{PSE}, & \text{if } \frac{1}{T}\sum_{t=1}^{T} P_t \geq \xi, \\ \text{FE}, & \text{otherwise.} \end{cases} \tag{10}$$

where $P_t$ is the predicted progress at frame $t$, $T$ is the trajectory length, and $\xi$ is the median of the average progress over the non-successful rollouts, ensuring an equal split between PSE and FE. By doing so, we avoid the bias introduced by manually setting thresholds for distinguishing between PSE and FE. We further compute a classification score $\rho$ by assigning $+1$ for each correct prediction and $-1$ for each incorrect one, normalized by the total number of rollouts (36), i.e., $\rho = \frac{\#\text{correct} - \#\text{wrong}}{36}$. We additionally report a breakdown of the estimation success rate (SR) for each category (SE, PSE, FE) across models, in order to highlight their distinct behaviors, such as being overly optimistic or assigning zero progress universally.

**Analysis on T-shirt folding task.** We prepare two datasets: $\mathcal{D}_{\text{dense}}$ with dense annotations containing 200 trajectories, and $\mathcal{D}_{\text{sparse}}$ with sparse annotations containing 500 trajectories, examples of dense and sparse annotated demonstration data can be found at Fig. 5 and 6. Importantly, these are distinct demonstrations rather than the same trajectories annotated differently. We reserve 10% of each dataset for testing and use the remainder for training. $\mathcal{D}_{\text{sparse}}$, due to its larger size, covers a wider range of scenarios, whereas $\mathcal{D}_{\text{dense}}$ provides more detailed per-trajectory labeling. We trained SARM reward model on both $\mathcal{D}_{\text{dense}}$ and $\mathcal{D}_{\text{sparse}}$, the details on model training are provided in Appendix A.4.

The detailed comparison results of the reward models are presented in Table 2. Among the baselines, **GVL**, **VLC**, and VICtoR failed to provide reliable reward signals: classification breakdowns reveal that **GVL** and VICtoR tend to be overly pessimistic, while **VLC** is excessively optimistic. **LIV,**

**ReWiND** , and REDS achieve stronger performance, delivering more accurate classifications of policy rollouts. However, due to the unstable reward labeling issues discussed in Section 3.1, their effectiveness remains limited on both human demonstrations and robot rollouts when compared with **SARM**. Although its regularization loss introduces a mild progressive trend, the estimation backbone of REDS remains step-shaped and semi-sparse, making it difficult to produce dense and accurate reward estimates.

For the ablation studies, both single-scheme variants underperform relative to our two-scheme model, highlighting the advantage of leveraging larger datasets with heterogeneous annotation protocols. Notably, the model trained solely on the dense annotation dataset performs poorly on unseen scenarios. Since this dataset is smaller and less diverse, it fails to capture the wide range of situations present in real-world rollouts, which often involve complex patterns such as back-and-forth motions, misgrasps, and recovery struggles. Training without rewind augmentation also results in degraded performance. While human demonstration evaluation remains largely unaffected—since the dataset does not contain deliberate failure cases and progress is generally monotonic—the performance on real robot rollouts drops substantially. In this case, the model becomes overly optimistic, failing to recognize regressions or failures. These findings demonstrate that rewind augmentation is essential for building reward models that generalize to real-world policies. SARM (VB) showcased SARM's robustness under varied lighting conditions, which is important when faced with diverse dataset.

In summary, our method consistently outperforms all baselines, achieving more than 50% relative improvement on human demonstration benchmarks and over 80% improvement on real robot rollouts compared to the strongest baseline, **ReWiND**.

**Analysis on dish unloading task.**  Dish unloading is a multi-stage task where the robot removes dishes from a rack and places them flat on a table. The process has three stages: (1) grasp and lift, (2) optionally hand over to the other arm, and (3) place on the table. Unlike T-shirt folding, it is shorter-horizon and does not involve deformable objects, but introduces greater execution diversity: varying dish counts, orientations, rack positions (left vs. right arm use), and whether handovers are needed. This variability makes reward modeling challenging, especially for visual understanding. We use a single dataset, $\mathcal{D}_{\text{dish}}$, with the same annotation protocol and training setup as T-shirt folding. As shown in Table 2, SARM consistently outperforms all baselines. The visualization results can be found at Fig. 13 and 14.

Table 2: Evaluation of reward models for "unloading dishes" task. "Demo $\mathcal{L}$" denotes the single-step MSE of reward models on the validation set. All models are evaluated on 30 trajectories which are not included in training set. Both ground-truth progress and model predictions are normalized to the $[0, 1]$ range. "Rollout $\rho$" reports performance on real policy rollouts.

| Metrics | GVL | VLC | LIV | ReWiND | w/o R | SARM |
|---|---|---|---|---|---|---|
| Demo $\mathcal{L}$ ↓ | 0.089 | 0.045 | 0.042 | 0.018 | **0.013** | **0.013** |
| Rollout $\rho$ ↑ | 0 | -0.33 | 0.39 | 0.55 | 0.50 | **0.67** |
| *Classification SR Breakdown* | | | | | | |
| SE | 0/12 | **12/12** | 7/12 | **12/12** | **12/12** | 10/12 |
| PSE | 6/12 | 0/12 | **12/12** | 9/12 | 7/12 | 9/12 |
| FE | **12/12** | 0/12 | 6/12 | 7/12 | 8/12 | 11/12 |

## 4.2  Q2: POLICY LEARNING WITH RA-BC

Folding a crumpled T-shirt is among the most challenging robotic manipulation tasks, as it requires robust visual understanding, long-horizon planning, and the ability to handle deformable objects. **Pi0** (Black et al.)  a RBM demonstrated the capabilities of completing this task. Although the policy weights have been open-sourced, the embodiment gap and the lack of high-quality datasets still makes it difficult to reproduce or further improve upon their results. For the remainder of this section, all policies we report are fine-tuned from **Pi0**.

**Dataset.**  We collect a large dataset $\mathcal{D}_{\text{all}}$ comprising 200 hours of T-shirt folding demonstrations using the GELLO teleoperation system (Wu et al., 2024) with YAM 7-DoF bimanual robotic arms.From this, we derive a smaller subset $\mathcal{D}_{\text{2min}}$ by filtering trajectories based on task duration,

Table 3: Success rates (SR) of T-shirt folding policies at 20K and 40K training steps. Each block reports the overall SR for each task. Detailed per-color results are provided in Table A.7.

| Training Steps | Tasks | (1) $\mathcal{D}_{\text{all}}$ | (2) $\mathcal{D}_{\text{2min}}$ | (3) ReWiND | (4) SARM |
|---|---|---|---|---|---|
| **20K** | Simple | **12/12** | **12/12** | **12/12** | **12/12** |
| | Medium | 0/12 | 4/12 | 1/12 | **7/12** |
| | Hard | 0/12 | 1/12 | 1/12 | **6/12** |
| **40K** | Simple | **12/12** | **12/12** | **12/12** | **12/12** |
| | Medium | 1/12 | 7/12 | 6/12 | **10/12** |
| | Hard | 0/12 | 0/12 | 3/12 | **8/12** |

retaining only those completed within 2 minutes, resulting in a 20-hour dataset. Each demonstration follows a structured procedure: (1) picking a randomly crumpled T-shirt from a box, (2) flattening the T-shirt, (3) folding the T-shirt, and (4) placing it neatly in the corner. To encourage generalization, we randomize T-shirt color and texture as well as the background environment. Aside from trajectory duration, however, no direct quantitative index is available to measure demonstration quality. Furthermore, annotations are not explicitly incorporated during policy training.

**Tasks and Evaluation Protocol.** To conduct a more detailed evaluation of the trained T-shirt folding policy, we decompose the task into three sub-tasks of increasing difficulty: (1) **Easy:** picking the shirt from the box and placing it at the center of the table, (2) **Medium:** folding the T-shirt starting from a flattened state, and (3) **Hard:** completing the full pipeline from a crumpled initial state. Task 1 is relatively simple, requiring only picking and placing skills, with human demonstrations typically lasting within 5 seconds. Task 2 demands contact-rich manipulation of deformable objects and long-horizon planning, with human demonstrations lasting 30 seconds to 1 minute. Task 3 is the most challenging, as it includes the flattening stage. Here, the robot must rely on strong visual understanding to handle occlusions and uncertainties inherent in deformable object manipulation. The policy must judge whether the T-shirt is sufficiently flattened, as failure to do so would compromise the subsequent folding step. Human demonstrations for Task 3 typically range from 1 to 3 minutes. For evaluation, we test each task using three different colored T-shirts (*red*, *black*, and *blue*). Each task is rolled out 4 times per color, for a total of 12 trials per task. Success criteria are defined as follows: for Task 1, the T-shirt must be picked from the box and placed at the table center within 1 minute; for Task 2, the T-shirt must be neatly folded and placed at the corner within 3 minutes; and for Task 3, the T-shirt must be neatly folded and placed at the corner within 5 minutes.

**Training Methods.** We fine-tune four policies in total: (1) **BC-All**, trained on the full dataset $\mathcal{D}_{\text{all}}$ using standard behavior cloning; (2) **BC-2min**, trained on the filtered high-quality subset $\mathcal{D}_{\text{2min}}$ using standard behavior cloning; (3) **RA-BC-ReWiND**, trained on $\mathcal{D}_{\text{all}}$ using RA-BC with a reward model trained by the ReWiND baseline (Zhang et al., 2025); and (4) **RA-BC-SARM**, trained on $\mathcal{D}_{\text{all}}$ using RA-BC with our proposed reward model, SARM. It is to be noted that (1) and (2) are trained *without* any reweighting. They serve as baselines/ablations to illustrate the performance of plain behavior cloning on the diverse dataset $\mathcal{D}_{\text{all}}$ and on the naively filtered subset $\mathcal{D}_{\text{2min}}$. The policy training details are listed in Appendix A.7. We evaluate policies at both 20k and 40k training steps and report their success rates for each task. The experiment results can be found at Table 4.2.

All policies achieve high success rates on the easy task (picking and placing the T-shirt), indicating that both the dataset and training procedure are sufficient for learning basic manipulation skills. On the medium task (folding from a flattened state), the policy trained on $\mathcal{D}_{\text{2min}}$ substantially outperforms the one trained on $\mathcal{D}_{\text{all}}$, with its success rate at 40k steps improving from near 0% to over 50%. This highlights the importance of carefully filtered, high-quality data for learning more complex manipulation behaviors. Nevertheless, this policy still fails on the hard task (folding from a crumpled state), indicating that filtering by duration alone is insufficient. Such a naive strategy cannot emphasize demonstrations that require advanced perception and decision making, such as judging whether the T-shirt has been adequately flattened before folding. As a result, it fails to deliver the dynamic and contact-rich manipulation needed for tasks that demand seamless coordination of both arms.

By leveraging SARM, the RA-BC policy surpasses both BC baselines by a significant margin on medium and hard tasks at both 20k and 40k steps. In particular, the RA-BC policy at 40k steps

achieves an 83% success rate on the medium task and a 67% success rate on the hard task. These results demonstrate that RA-BC effectively exploits diverse datasets by filtering high-quality data frames, enabling the policy to learn robust long-horizon manipulation strategies.

### 4.3 Q3: Effect of Reward Model Quality in RA-BC

To investigate how the quality of the reward model influences RA-BC performance, we conduct an ablation study by training the T-shirt folding policy using RA-BC with the baseline **ReWiND** (Zhang et al., 2025) reward model. The evaluation results of reward models are presented in Table 2, and the corresponding policy performance is summarized in Table 4.2. Compared to RA-BC with SARM, RA-BC with the ReWiND reward model achieves substantially lower success rates on both the medium (83% v.s. 50%) and hard tasks (67% v.s. 25%). In particular, on the medium task, its performance drops to the level of the vanilla BC baseline trained on the filtered dataset $\mathcal{D}_{2\min}$.

These results highlight the central role of reward model quality in RA-BC. A reliable model accurately captures task progress, enabling effective filtering of demonstrations and consistent supervision for policy learning. In contrast, a poor model misjudges progress, misweights data, and weakens the benefits of filtering. This reliability is especially crucial in long-horizon, multi-stage tasks like T-shirt folding, where failures and partial progress are common.

We further explore the use of SARM in reinforcement learning (RL) by training a DiffQL (Wang et al., 2022) manipulation policy with reward signals provided by SARM. The detailed methodology, experimental setup, and results are presented in Appendix A.8.

## 5 Conclusion

This paper explored how demonstration quality shapes the effectiveness of RBMs when tackling complex, long-horizon manipulation. Using T-shirt folding as a demanding case study, we showed that naïvely scaling dataset size is insufficient, and that progress-aware supervision is needed to guide learning. To this end, we designed SARM, a stage-aware, video-based reward modeling framework that transforms natural language annotations into structured progress signals, enabling more reliable estimation of task advancement across diverse demonstrations. We further introduced the RA-BC framework, which incorporates these signals to emphasize higher-value trajectories during training. Our empirical evaluation revealed clear benefits: SARM consistently surpassed prior baselines, and policies trained with RA-BC achieved strong performance on real robots, including an 83% success rate when folding T-shirts from a flattened configuration and 67% when starting from a crumpled state. Additional analysis showed that the accuracy of the reward model is pivotal—when the reward signal is weak, RA-BC loses its ability to properly weight samples and overall policy performance degrades. We also demonstrate that SARM can be incorporated into reinforcement learning (RL) to further improve policy performance by modifying DiffQL Wang et al. (2022) on a pick-and-place task in the MuJoCo Todorov et al. (2012) simulation environment. Details are provided in A.8. These findings underscore that high-quality reward modeling, combined with selective data filtering, is a powerful path forward for building robust and scalable RBMs capable of addressing long-horizon manipulation challenges.

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

## A APPENDIX

### A.1 LLM USAGE DISCLOSURE.

In accordance with the submission policy, we disclose that large language models (LLMs) were used only for minor wording improvements, grammar checking, and proofreading. All ideas, technical content, experimental design, and analysis were solely developed by the authors, who take full responsibility for the final manuscript.

### A.2 HARDWARE SETUP

For our real world experiments we leverage a bimanual robot table top platform. The system consists of:

- Two 6 DOF YAM robot arms, built by the manufacturer I2RT (I2RT-Robotics, 2025).
- Three RealSense D405 cameras. One for each wrist and a third statically mounted above for viewing the scene (RealSense, 2025).

Data is collected using a leader follower system GELLO teleoperation system(Wu et al., 2024). The environment run at recorded at 30 fps and includes synchronized streams from three cameras (left wrist, right wrist, and a fixed top view) along with robot joint angles and action joint angle commands.

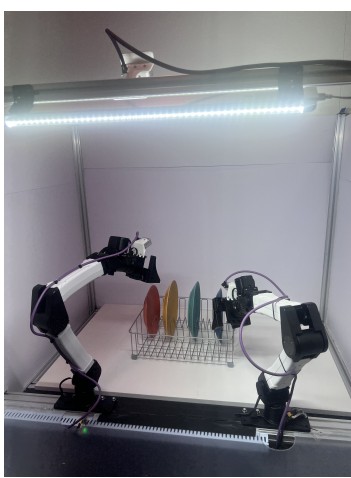

Figure 4: The physical station used for data collection and policy evaluation.

### A.3 T-SHIRT FOLDING EXPERT DEMONSTRATION DATA

We provide two examples of demonstration trajectories collected with the **GELLO** system in Fig. 5 and Fig. 6. For clarity, the annotation corresponding to each motion stage is shown above every frame. Compared to $\mathcal{D}_{\text{sparse}}$, the dense annotated dataset $\mathcal{D}_{\text{dense}}$ further decomposes the overall "fold the T-shirt" stage into five fine-grained motions. A comparison of the average temporal portion of each task in the two datasets is summarized in Table A.3. It is important to note that $\mathcal{D}_{\text{sparse}}$ and $\mathcal{D}_{\text{dense}}$ are distinct datasets collected from different trajectories; therefore, even for the same task (e.g., "flatten the T-shirt out"), the average temporal portion differs a little across datasets.

Table 4: Average temporal portion of each task in two dataset.

| Sparse Annotated Dataset $\mathcal{D}_{\text{sparse}}$ | | Dense Annotated Dataset $\mathcal{D}_{\text{dense}}$ | |
|---|---|---|---|
| Task | Portion (%) | Task | Portion (%) |
| Grab the T-shirt from the pile | 5 | Grab T-shirt and move to center | 9 |
| Move the T-shirt to the center | 5 | Flatten out the T-shirt | 26 |
| Flatten the T-shirt out | 25 | Grab near side and fold | 15 |
| Fold the T-shirt | 55 | Grab far side and fold | 13 |
| Put folded T-shirt into corner | 10 | Rotate the T-shirt 90 deg | 8 |
| | | Grab bottom and fold | 9 |
| | | Grab 2/3 side and fold | 9 |
| | | Put folded T-shirt into corner | 11 |

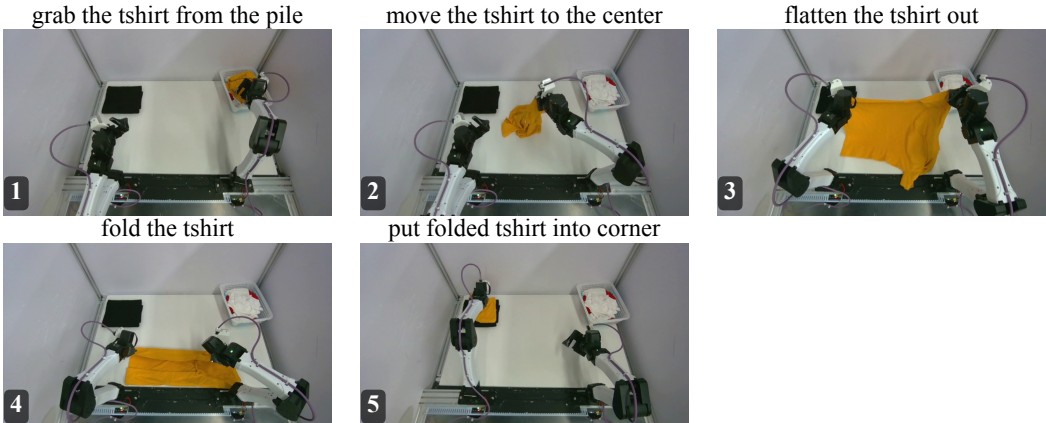

Figure 5: Expert demonstration with sparse annotation.

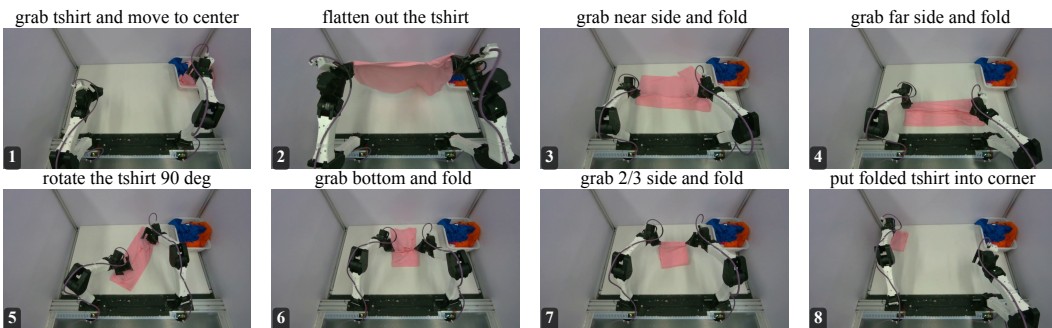

Figure 6: Expert demonstration with dense annotation.

### A.4 Reward Model Training

**Implementation Details.** We employ a frozen `clip-vit-base-patch32` encoder to process both RGB image sequences and task descriptions. The dataset is recorded at a fixed frame rate of 30 fps, and each input sequence consists of 9 images: the first is always the initial frame of the episode, while the remaining 8 are sampled consecutively from the same episode with a fixed interval of 30 frames, resulting in a temporal span of approximately 8 seconds. To enhance temporal diversity and better capture failure scenarios, we follow the rewind augmentation strategy (Zhang et al., 2025), appending up to four frames from earlier timestamps with reversed order to the end of each training sequence. Additionally, to improve video-language alignment, the task descriptions are occasionally perturbed with randomly generated incorrect instructions.

The backbone is a transformer-based temporal aggregator with 8 layers, 12 attention heads, and a hidden dimension of 768. To mitigate information leakage, positional embeddings are applied only to the first frame, corresponding to the episode start. On top of the backbone, we incorporate twin MLP-based output heads tailored for different annotation types, namely dense and sparse labels. This design enhances the flexibility of the reward model, allowing it to effectively utilize heterogeneous supervision and remain compatible with multiple annotation protocols. Each output head comprises 2 layers with a hidden dimension of 512. The stage model is trained with cross-entropy loss, whereas the subtask model is optimized with mean squared error loss.

Optimization is performed with the AdamW optimizer, using a learning rate of $5 \times 10^{-5}$ and a weight decay of $1 \times 10^{-3}$. Models are trained for 2 epochs with a batch size of 64 on a single NVIDIA RTX 4090 GPU.

**Scale Analysis.** We study the effect of model scale on reward model performance by varying the number of transformer layers in the temporal aggregator from 4, 8, to 12, corresponding to models with 30M, 60M, and 90M parameters, respectively, while keeping all other hyperparameters fixed. The results are summarized in Table 5 and Fig. 7. The smallest model (30M) exhibits clear underfitting, with poor performance across both evaluation metrics. Increasing the size from 30M to 60M leads to substantial gains, but further scaling from 60M to 90M yields only marginal or negligible improvement. This suggests that a 60M-parameter model is sufficient to capture the task dynamics of T-shirt folding. Larger models risk overfitting, particularly given the limited size of the training data. Overall, the chosen 60M configuration strikes an effective balance between model capacity and computational efficiency, and is therefore adopted throughout the paper.

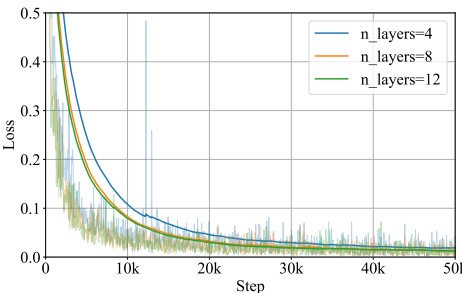

Figure 7: Scale analysis plots of reward models with various layers.

Table 5: Scalability analysis of reward model on T-shirt folding task

| Metrics | Layer Number | | |
|---|---|---|---|
| | 4 | 8 | 12 |
| Demo $\mathcal{L} \downarrow$ | 0.015 | 0.009 | **0.007** |
| Rollout $\rho \uparrow$ | 0.72 | **0.94** | 0.88 |
| *Classification SR Breakdown* | | | |
| SE | 10/12 | **12/12** | 12/12 |
| PSE | 10/12 | **11/12** | 11/12 |
| FE | 11/12 | **12/12** | 11/12 |

**Ablation Study.** We perform a comprehensive ablation study to evaluate the impact of different design choices in reward model training. Specifically, we examine: (1) the use of joint state as an additional input, (2) the inclusion of wrist cameras, (3) the number of observation steps, and (4) the frame gap between observation steps. The corresponding training loss curves are shown in Fig. 8.

*Joint state input:* We compare two variants of the reward model: one that incorporates the robot's joint state (SARM's default configuration) and one that relies only on visual observations. As shown in Table 6, including joint state leads to more accurate estimation on both human demonstrations and policy rollouts. The improvement, however, is relatively modest, suggesting that visual input already contains most of the task-relevant information.

*Wrist cameras:* We evaluate the effect of adding wrist camera views in addition to the fixed top-down camera. The results in Table 7 show little to no benefit, likely because the top-down perspective already provides sufficient task coverage, while wrist cameras contribute redundant information. Moreover, incorporating wrist views increases system complexity and introduces a threefold I/O cost. For these reasons, we exclude them from the final design.

*Number of observation steps:* We vary the number of observation steps from 4, 8 (SARM's default), to 12, while keeping the temporal horizon fixed at approximately 8 seconds. Results in Table 8 indicate that too few steps (4) limit the model's ability to capture temporal dynamics, resulting in underfitting. On the other hand, too many steps (12) introduce redundancy and additional computational overhead without clear gains. An intermediate choice of 8 steps provides a good balance between temporal coverage and efficiency.

*Frame gap between steps:* We also assess the effect of varying the frame gap between consecutive observation steps at 15, 30 (SARM's default), and 60 frames. As shown in Table 9, a small gap of 15 frames (0.5s) produces highly correlated inputs, limiting temporal diversity and shortening the effective temporal span, which leads to underfitting. A large gap of 60 frames (2s) risks missing important intermediate states, thereby confusing the model and degrading performance. A moderate gap of 30 frames (1s) achieves the best trade-off by capturing meaningful temporal transitions while avoiding redundancy.

In summary, these results underscore the importance of balancing model capacity, input modalities, and temporal resolution. Our final design choices—using joint state input, excluding wrist cameras, adopting 8 observation steps, and a frame gap of 30 frames—reflect the insights gained from this ablation analysis and provide a robust configuration for long-horizon reward modeling.

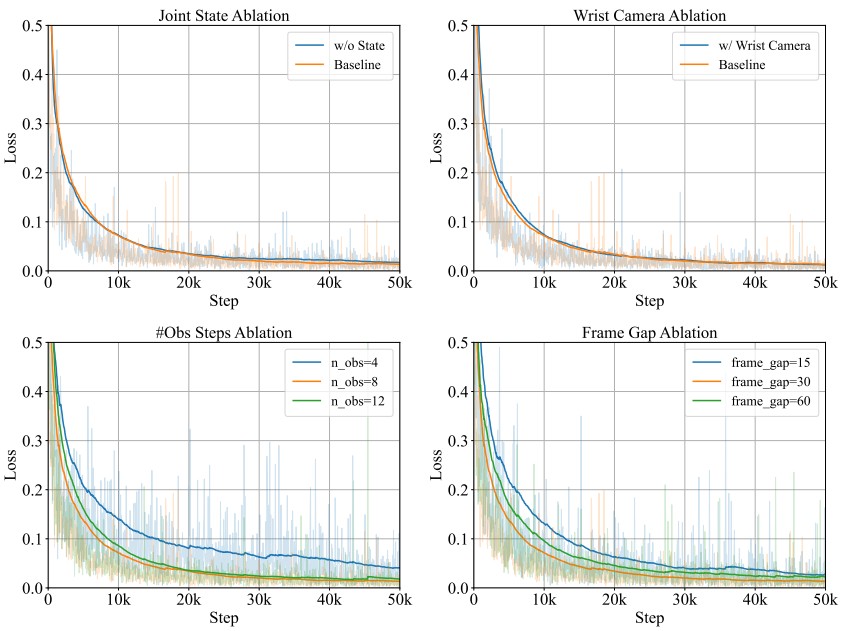

Figure 8: Ablation study training loss curves

Table 6: Ablation study of T-shirt folding reward model on using joint state (our choice: Yes).

| Metrics | Using Joint State (our choice: Yes). | |
| --- | --- | --- |
| | Yes | No |
| Demo $\mathcal{L} \downarrow$ | 0.010 | **0.009** |
| Rollout $\rho \uparrow$ | 0.72 | **0.94** |
| *Classification SR Breakdown* | | |
| SE | **12/12** | 12/12 |
| PSE | 9/12 | **11/12** |
| FE | 10/12 | **12/12** |

Table 7: Ablation study of T-shirt folding reward model on using wrist cameras (our choice: No).

| Metrics | Using Wrist Cameras | |
| --- | --- | --- |
| | Yes | No |
| Demo $\mathcal{L} \downarrow$ | **0.008** | 0.009 |
| Rollout $\rho \uparrow$ | **0.94** | **0.94** |
| *Classification SR Breakdown* | | |
| SE | **12/12** | **12/12** |
| PSE | **11/12** | **11/12** |
| FE | **12/12** | **12/12** |

Table 8: Ablation study of T-shirt folding reward model on observation steps number (our choice: 8).

| Metrics | Observation Step Number | | |
| --- | --- | --- | --- |
| | 4 | 8 | 12 |
| Demo $\mathcal{L} \downarrow$ | 0.013 | **0.009** | **0.009** |
| Rollout $\rho \uparrow$ | 0.67 | **0.94** | 0.89 |
| *Classification SR Breakdown* | | | |
| SE | 12/12 | **12/12** | **12/12** |
| PSE | 8/12 | **11/12** | **11/12** |
| FE | 10/12 | **12/12** | 11/12 |

Table 9: Ablation study of T-shirt folding reward model on sequence frames gap (our choice: 30).

| Metrics | Frame Gap | | |
| --- | --- | --- | --- |
| | 15 | 30 | 60 |
| Demo $\mathcal{L} \downarrow$ | 0.015 | **0.009** | 0.022 |
| Rollout $\rho \uparrow$ | 0.50 | **0.94** | 0.56 |
| *Classification SR Breakdown* | | | |
| SE | 9/12 | **12/12** | **12/12** |
| PSE | 8/12 | **11/12** | 8/12 |
| FE | 10/12 | **12/12** | 8/12 |

## A.5 REWARD MODEL EVALUATION RESULTS VISUALIZATION

**Demo Data Estimation.** We present two visualization examples of reward predictions from **SARM** and the **ReWiND** baseline in Fig. 9 and Fig. 10, using trajectories from the validation set of human demonstration data. Compared with SARM, ReWiND exhibits several notable shortcomings: (1) as it relies solely on direct regression, it fails to capture the full progression of long-horizon tasks—for instance, its predictions do not start at zero even at the beginning of a trajectory; and (2) its estimates are highly unstable, with frequent oscillations and even negative spikes, which should not occur in human demonstration data. These issues prevent ReWiND from producing consistent long-horizon reward signals and ultimately limit its effectiveness for downstream policy learning.

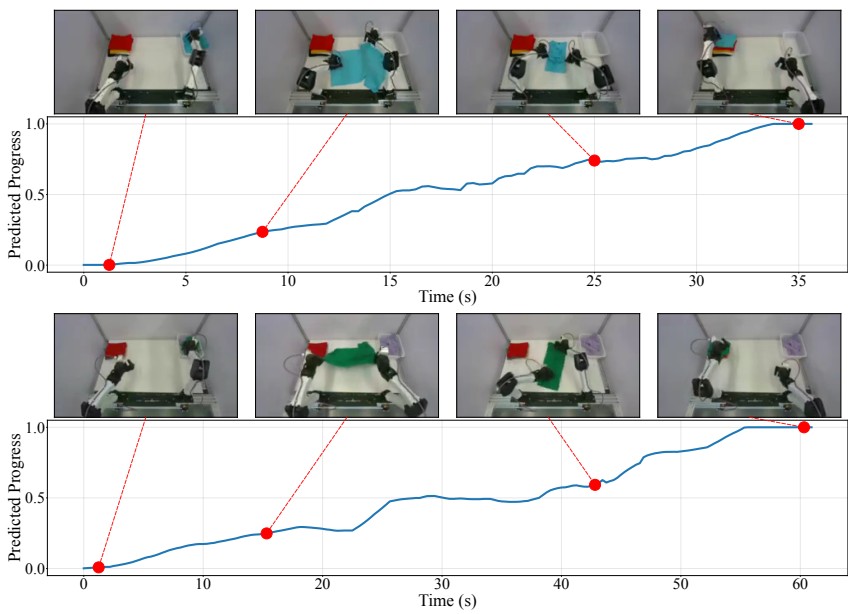

Figure 9: Examples of proposed reward model prediction on demonstration data.

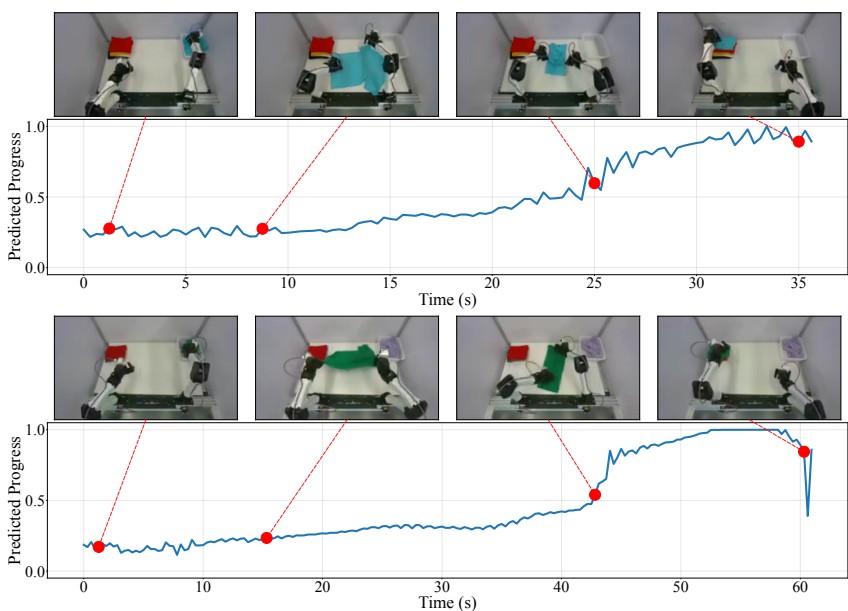

Figure 10: Examples of ReWiND reward model prediction on demonstration data.

**Policy Rollout Estimation.** We present two visualization examples of reward predictions from **SARM** and the **ReWiND** baseline in Fig. 11 and Fig. 12, using trajectories from real robot policy rollouts. Compared with human demonstration data, policy rollouts are more challenging because they often include failure modes that are out-of-distribution (OOD), such as misgrasps, recovery attempts, and back-and-forth motions. In the first example, the trajectory corresponds to a successful rollout where the robot folds the T-shirt correctly, with only minor struggles and misgrasps in the first ten seconds. In this case, SARM remains stable, keeping the estimated progress near zero during these OOD motions, whereas ReWiND is easily triggered and produces noisy, unstable estimates. The second example highlights a failed rollout, with four key frames: (1) the T-shirt is flattened after struggling, (2) folding is nearly complete, (3) the robot suddenly fails and crumples the T-shirt on the table, and (4) the unfolded T-shirt is placed in the corner. SARM provides reasonable progress estimates across all four stages, reflecting the actual task status. By contrast, ReWiND continues to exhibit high noise and spurious spikes, and even assigns a high progress score to the final "fake finish" state, effectively being misled by the failed outcome. These results further emphasize the robustness and reliability of SARM framework for real-world robotic applications.

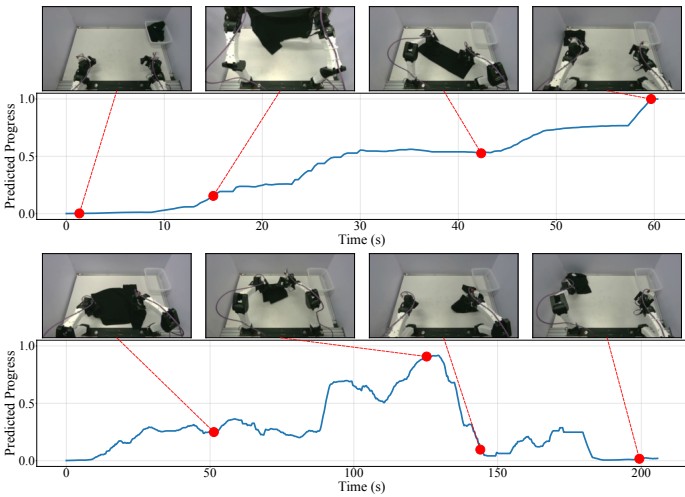

Figure 11: Examples of proposed reward model prediction on policy rollouts.

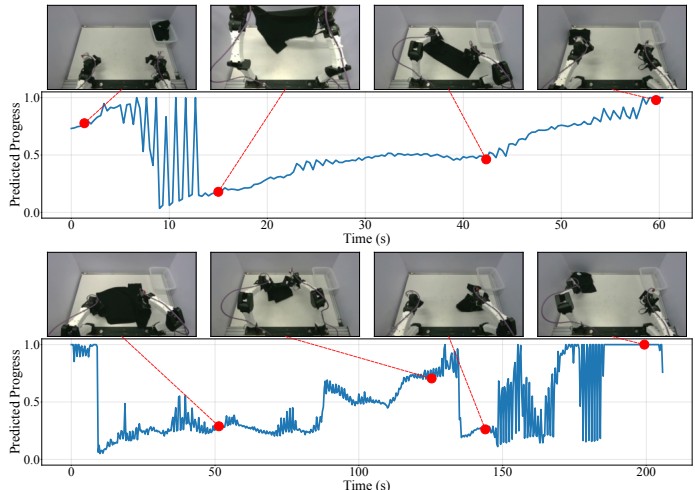

Figure 12: Examples of ReWiND reward model prediction on policy rollouts.

## A.6 TRAINING SARM FOR DISH UNLOADING

**Demo Data Estimation.** The visualization of reward predictions from **SARM** and the **ReWiND** baseline on two example trajectories from the validation set of human demonstration data is shown in Fig. 13 and Fig. 14. SARM produces consistent and robust progress estimates, maintaining stable predictions for sequences as long as unloading eight dishes consecutively, which corresponds to over 1.5 minutes of execution. This demonstrates the effectiveness of SARM in handling diverse and highly dynamic tasks. By contrast, ReWiND exhibits similar shortcomings as in the T-shirt folding experiments: it fails to capture the full progression of the task (in this case never estimating completion, with peak values below 0.75) and generates unstable predictions with noticeable fluctuations.

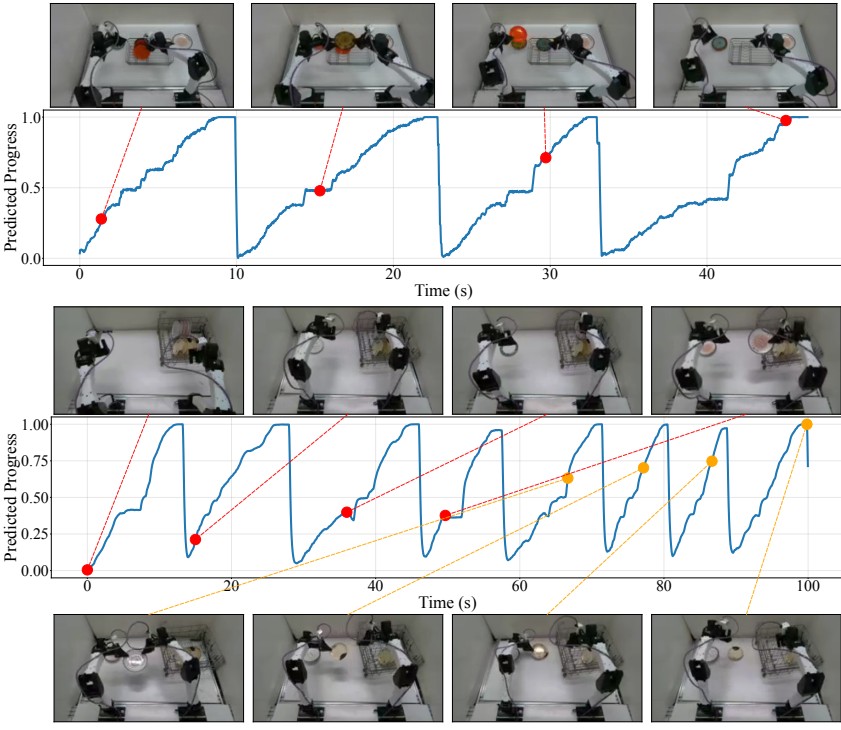

Figure 13: Examples of proposed reward model prediction on demonstration data.

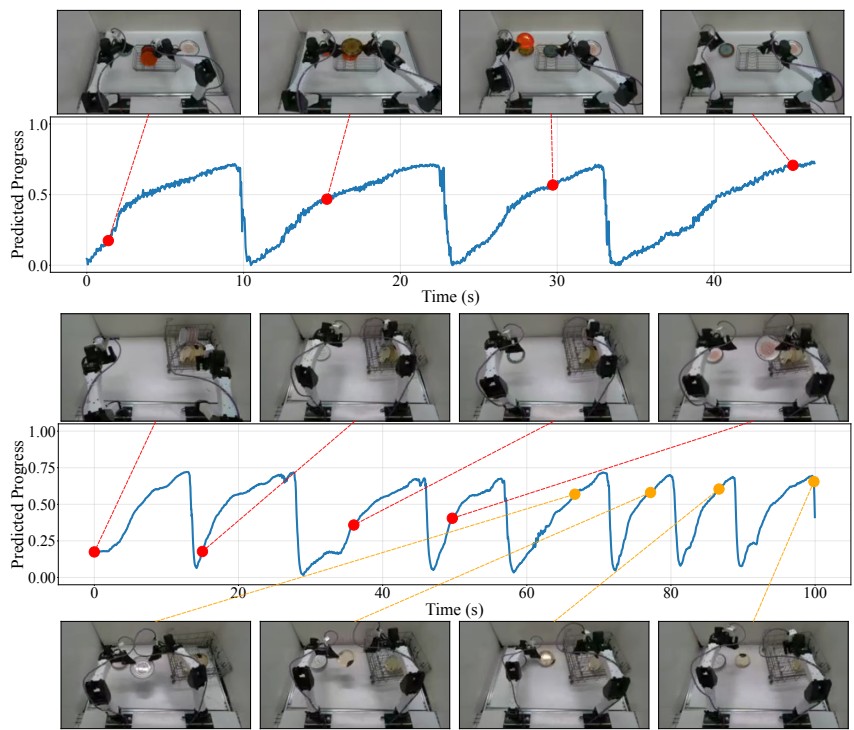

Figure 14: Examples of ReWind reward model prediction on demonstration data.

### A.7 TRAINING MANIPULATION POLICY WITH RA-BC

**Implementation Detail.** All policies are fine-tuned with low rank adaptation (LoRA) (Hu et al., 2022) for 40k steps using a batch size of 32 on a dual NVIDIA RTX 4090 machine. The RA-BC hyperparameters are set to $\kappa = 0.01$ and $\epsilon = 10^{-6}$, with a chunk length of $\Delta = 25$ actions to align with the policy's action chunking. Since the dataset is recorded at 30 fps, $\kappa = 0.01$ corresponds roughly to a task duration of 1 minute 30 seconds, which represents the threshold for the top 5% of demonstrations. For data points better than this threshold, we assign a weight of 1. For data points that are worse than this threshold but still demonstrate positive progress, we assign a soft weight between 0 and 1 according to Eq. 8. For data points exhibiting negative progress, the assigned weight is 0.

**Loss Curve.** The policy training loss curves for all four methods are shown in Fig. 15. We observe that the two pure BC methods initially exhibit a faster decrease in loss compared to RA-BC, but they plateau early and converge to a higher final loss value. In contrast, RA-BC methods display a slower but more consistent reduction in loss and ultimately achieve lower convergence values. This phenomenon can be explained by the data distribution: pure BC leverages a broader set of demonstrations, which allows an unconverged policy to quickly match parts of the dataset, resulting in smaller loss at the early stages of training. However, this diversity also introduces conflicting gradient signals that prevent further improvement, causing convergence at a suboptimal plateau. On the other hand, RA-BC employs a more focused learning objective that emphasizes high-quality data. Although such data is harder to fit initially, the targeted supervision enables the policy to continue improving and avoid stagnation. The final converged loss values are consistent with the policy evaluation results in Table A.7, where RA-BC with our SARM framework delivers the best overall performance.

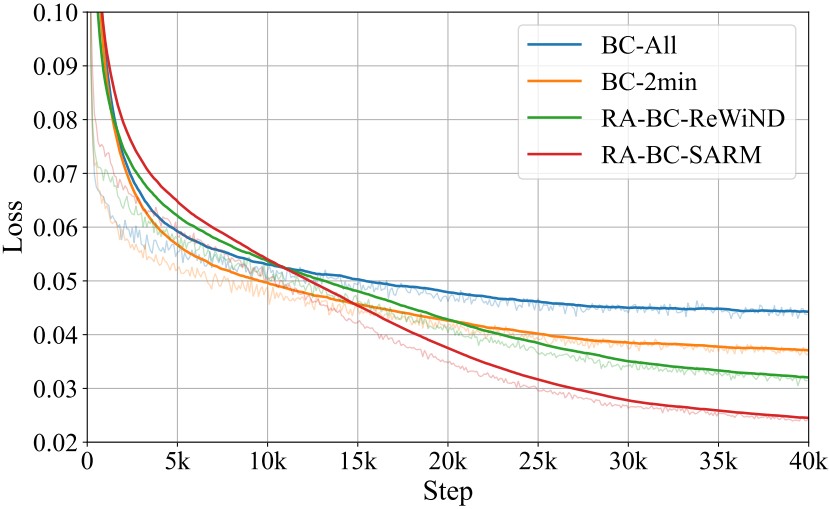

Figure 15: Trained T-shirt folding policies loss curves.

Table 10: Success rates (SR) of T-shirt folding policies at 20K and 40K training steps. For each method, the first column reports SR for each T-shirt color (**R** = red, **B** = blue, **K** = black), and the last row in each block shows the overall SR for that task.

| Training Steps | Tasks | Color | (1) $\mathcal{D}_{all}$ | (2) $\mathcal{D}_{2min}$ | (3) ReWiND | (4) SARM |
|---|---|---|---|---|---|---|
| | | R | 4/4 | 4/4 | 4/4 | 4/4 |
| | | B | 4/4 | 4/4 | 4/4 | 4/4 |
| | Simple | K | 4/4 | 4/4 | 4/4 | 4/4 |
| | | **Overall** | **12/12** | **12/12** | **12/12** | **12/12** |
| | | R | 0/4 | 3/4 | 0/4 | 3/4 |
| **20K** | | B | 0/4 | 1/4 | 1/4 | 2/4 |
| | Medium | K | 0/4 | 0/4 | 0/4 | 2/4 |
| | | **Overall** | **0/12** | **4/12** | **1/12** | **7/12** |
| | | R | 0/4 | 1/4 | 1/4 | 2/4 |
| | | B | 0/4 | 0/4 | 0/4 | 4/4 |
| | Hard | K | 0/4 | 0/4 | 0/4 | 0/4 |
| | | **Overall** | **0/12** | **1/12** | **1/12** | **6/12** |
| | | R | 4/4 | 4/4 | 4/4 | 4/4 |
| | | B | 4/4 | 4/4 | 4/4 | 4/4 |
| | Simple | K | 4/4 | 4/4 | 4/4 | 4/4 |
| | | **Overall** | **12/12** | **12/12** | **12/12** | **12/12** |
| | | R | 0/4 | 4/4 | 2/4 | 4/4 |
| **40K** | | B | 1/4 | 1/4 | 3/4 | 4/4 |
| | Medium | K | 0/4 | 2/4 | 1/4 | 2/4 |
| | | **Overall** | **1/12** | **7/12** | **6/12** | **10/12** |
| | | R | 0/4 | 0/4 | 2/4 | 2/4 |
| | | B | 0/4 | 0/4 | 0/4 | 4/4 |
| | Hard | K | 0/4 | 0/4 | 1/4 | 2/4 |
| | | **Overall** | **0/12** | **0/12** | **3/12** | **8/12** |

**Rollout Example.** An example policy rollout is shown in Fig. 16, where the robot successfully folds a T-shirt from a crumpled state into a neat configuration within 90 seconds, without any misgrasps.

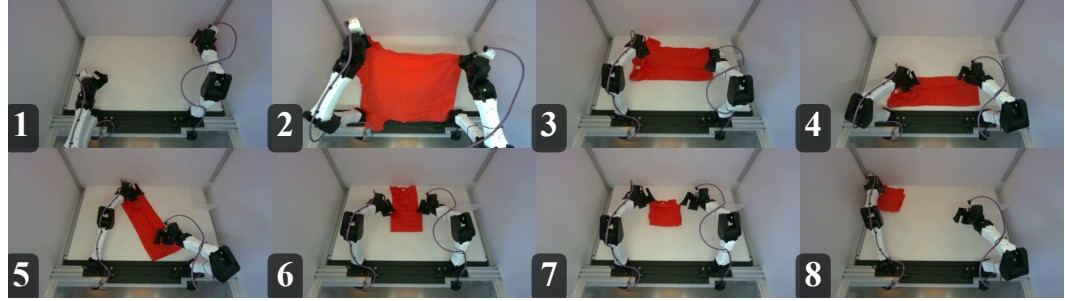

Figure 16: Example of RA-BC trained T-shirt folding policy rollout.

A.8 REINFORCEMENT LEARNING EXAMPLE.

Beyond RA-BC, we also explored integrating SARM with reinforcement learning (RL) to further improve policy performance. We adopt a two-stage training scheme: *(1) pre-training*, where the policy is trained with pure behavior cloning (BC) using the diffusion policy (Chi et al., 2023); and *(2) fine-tuning*, where the policy is refined with DiffQL (Wang et al., 2022), a Q-learning method specifically designed for diffusion-based policies. We refer to this approach as **RA-QL**.

We evaluate RA-QL on a simple two-stage task: pick up a cube. In this task, the robot arm must first reach toward the cube on the desktop, then grasp and lift it into a goal region, which is a fixed designated box space. An illustration of the task is provided in Fig. 17. We collected 300 expert demonstrations in the MuJoCo simulation environment (Todorov et al., 2012) with randomized cube initial position, denoted as $\mathcal{D}_{cube}$. From these, 100 trajectories were annotated to train a reward model with the same architecture used for T-shirt folding.

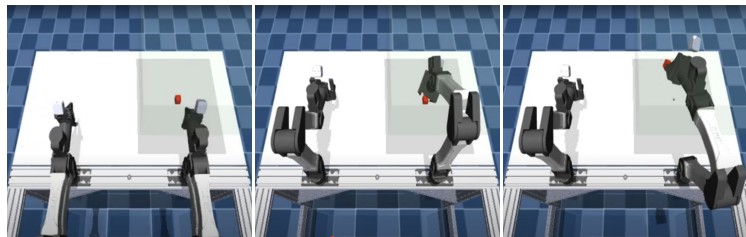

Figure 17: Expert demonstration of the cube-picking task. The desired goal region is highlighted in green.

We first pre-trained a diffusion transformer policy on $\mathcal{D}_{cube}$ using pure BC. While the policy achieved high success rates, the motions were often inefficient and imprecise, especially during the reaching and grasping phases. We then fine-tuned this BC-trained policy with DiffQL for an additional 10k steps. Our implementation closely follows DiffQL (Wang et al., 2022), with the key modification that the critic network receives rewards from our learned reward model, SARM, rather than hand-crafted signals. For ablation, we also continued training the diffusion policy with pure BC for another 10k steps under the same conditions.

During fine-tuning, we evaluated both the BC and RA-QL policies every 500 steps. Each policy was rolled out 10 times with randomized cube positions, and we report the average success rate (SR) and average discounted return. The experiment results are demonstrated in Fig. 18 The reward function is automatically judged by the simulator: a step reward of 1 is given if the cube is lifted to the desired height, and 0 otherwise. An episode terminates either when the cube is successfully placed in the goal region (labeled as `success`) or when the horizon of 1000 steps is reached (labeled as `fail`).

$$G_t = \sum_{k=0}^{T-t-1} \gamma^k r_{t+k}, \tag{11}$$

where $G_t$ denotes the discounted return from time step $t$ and $\gamma = 0.995$ is the discount factor.

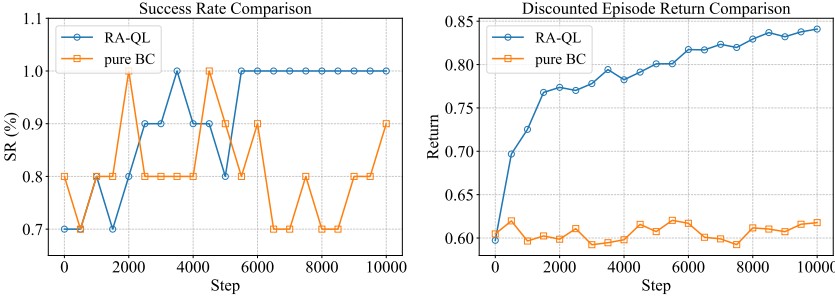

Figure 18: Comparison of RA-QL and pure BC on picking up cube policy training.

