# OpenReview forum: "SARM: Stage-Aware Reward Modeling for Long Horizon Robot Manipulation"
_ICLR.cc/2026/Conference — ICLR 2026 Poster_

### Official Review · Reviewer_WSRa · 2025-10-23

**Soundness:** 2
**Presentation:** 2
**Contribution:** 1
**Rating:** 2
**Confidence:** 4

**Summary:**

The paper proposes a reward-modeling approach that supplies stable, dense feedback for long-horizon manipulation. It decomposes progress estimation into two parts: (1) a high-level predictor that identifies the current stage of the task and (2) a fine-grained regressor that measures progress within that stage. This two-stage design reduces sensitivity to variable demonstration lengths (e.g., in cloth folding) and yields more reliable rewards. Building on this signal, the authors also introduce an imitation-learning scheme that up-weights higher-quality demonstrations. Experiments show consistent gains over single-stage progress-prediction baselines, with real-robot results on cloth-folding validating the approach.

**Strengths:**

- Strong real-world gains: On a challenging cloth-folding task, a naive fine-tuning SOTA behavior policy (Pi0) fails 0/12 trials, while the proposed method succeeds in 8/12—highlighting the efficacy of their good reward modeling.
- The paper analyzes performance across (i) high-level stage-prediction accuracy, and (ii) end-to-end policy rollouts, helping readers to understand where the method adds value.
- Consistent improvements: The performance improvement holds across different rollout outcomes (success/partial/failure progress prediction) and task difficulty, from simple cloth pick-and-place to full folding.

**Weaknesses:**

**Novelty of the stage-aware reward modeling.**
  + Stage-aware reward modeling for long-horizon manipulation is not new per se. Please clarify what is distinct, and compare directly to prior work (e.g., Drs [1], REDS [2]).

**Clarity on data-quality literature.**

  + L76–79 state data quality is hard to assess beyond simple heuristics (e.g., duration). This could be misleading: recent work proposes stronger proxies. Please cover this literature (like [3] and [4]) and briefly position your method relative to these proxies.

Please clarify the literature on stage-aware reward modeling and data quality, state how your approach differs from prior work, and—if possible—add experiments showing superiority over these to strengthen your paper’s contribution and rigor.

### References
- [1] Mu et al., "Drs: Learning reusable dense rewards for multi-stage tasks", ICLR 2024.
- [2] Kim et al., "Subtask-Aware Visual Reward Learning from Segmented Demonstrations", ICLR 2025.
- [3] Belkhale et al., "Data Quality in Imitation Learning", NeurIPS 2023
- [4] Dass et al., "DataMIL: Selecting Data for Robot Imitation Learning with Datamodels", arXiv preprint arXiv:2505.09603 2025

**Questions:**

- [Q1] Is the low performance due to underfitting?
    + In Table 2, ReWiND improves with longer training (e.g., 20K Medium: 1/12 → 40K Medium: 6/12). Does this indicate the model hasn’t converged yet and could achieve higher performance with more training? (possibly outperform your model as well?)

- [Q2] Model architecture parity
    + Are ReWiND and SARM matched in model size? Since original ReWiND leverages large-scale data (OXE), it may use a larger model, which could overfit on your 200-hour dataset.

---

> ### Author Response · Authors · 2025-11-20
> **Response to Reviewer WSRa (1/2)**
>
> Dear Reviewer WSRa,
>
>
> Thank you for your detailed and insightful comments. We address each of your questions below.
>
>
> ## (W1) Novelty and comparison to prior stage-aware reward models
>
>
> We thank the reviewer for raising the question regarding novelty in stage-aware reward modeling. We have added both DrS [1] and REDS [2] to our related work and included experimental results for REDS in Table 1.
>
>
> We would like to clarify that SARM is substantially different from both DrS and REDS in formulation, supervision, and architectural design:
>
>
> 1. DrS (ICLR 2024)\
> DrS is state-based, requiring full simulator proprioceptive states to determine stage indicators.
> It also requires training a separate discriminator for each stage, making it difficult to scale to long-horizon tasks with many subtasks.
> In contrast, SARM is fully video-based and performs stage estimation and progress regression directly from raw visual observations, without relying on simulator states or stage-specific models.
> Therefore, we have not added DrS experiments, because a direct comparison is difficult and offers limited insight: DrS relies on simulator-state supervision that is unavailable in our real trajectories, and its multi-discriminator design does not scale to our long-horizon visual setting.
>
>
> 2. REDS (ICLR 2025)\
> REDS relies on a semi-sparse, step-shaped reward with monotonicity regularization; it does not directly model dense progress.
> REDS does not have a dedicated stage-estimation module; instead, it infers stages via image–subtask embedding similarity, which we have found to be unreliable in practice.
> SARM explicitly models stage structure and predicts a dense frame-wise progress curve, which is crucial for long-horizon, contact-rich manipulation.
>
>
> As shown in Table 1 (Column 4) as well as the below, REDS underperforms both SARM and our strongest baseline, ReWiND. We also observed that REDS’s regularization introduces only a mild progressive trend, while its predictions remain step-shaped, limiting its ability to produce accurate dense reward estimates.
>
>
> | Metrics          | REDS   | ReWiND | SARM  |
> |------------------|--------|--------|-------|
> | Demo Loss         | 0.036  | 0.019  | 0.009 |
> | Rollout Score      | 0.16   | 0.50   | 0.94  |
> | Classification SR Breakdown |
> | SE               | 0/12   | 12/12  | 12/12 |
> | PSE              | 12/12  | 8/12   | 11/12 |
> | FE               | 7/12   | 7/12   | 12/12 |
>
>
> ## (W2) Clarity on data-quality proxies
>
>
> We thank the reviewer for pointing out the importance of positioning our work relative to recent advances in data-quality assessment. In the revised version, we have cited both Belkhale et al. [3] and Dass et al. [4], and updated L51–53 to clarify that although more sophisticated data-modeling approaches exist, lightweight, per-trajectory proxies remain challenging in practice.
>
>
> More specifically:\
> [3] defines data quality via the distribution shift between a learned policy's state-visitation distribution and that of an expert. This requires:
> - access to a reasonably strong expert policy
> - the ability to train policies in the environment
> - an IL algorithm–specific formulation
>
> These requirements are difficult to satisfy precisely when high-quality demonstrations are lacking—the scenario where reward modeling is most needed.
>
>
> [4] is concurrent work using influence functions for data selection; it also requires nontrivial training pipelines rather than simple per-trajectory heuristics.
>
>
> Both methods require learning + policy rollouts, fundamentally different from the simple duration heuristics we were referring to. We revised the paper accordingly.
>
>
> ## (Q1) Is low performance due to underfitting?
>
>
> We thank the reviewer for this important question. In the revised version, we added the policy training loss curves in Fig. 15. These results show that both: RA-BC with SARM, and RA-BC with ReWiND reach a clear training plateau by 40k steps, indicating convergence. Therefore, the performance gap is not due to underfitting, and we do not expect additional training to substantially improve ReWiND performance or surpass SARM.
>
>
> ## (Q2) Model architecture parity
>
>
> We thank the reviewer for the question regarding matched model capacity. SARM and ReWiND use identical backbone sizes for the transformer-based temporal aggregation modules. We have added a clarifying statement in Sec. 4.1 (L321–322):
> Both reward models use an 8-layer transformer,
> 12 attention heads,
> hidden dimension = 768.
>
>
> The original ReWiND implementation used a 4-layer transformer, but as shown in Fig. 7 and Table 4, such a small model underfits on our 200-hour dataset. Therefore, matching the model capacity for both methods provides a fair and stable comparison.
>
>
> Sincerely,\
> The Authors

---

> > ### Author Response · Authors · 2025-11-20
> > **Response to Reviewer WSRa (2/2)**
> >
> > References\
> > [1] Mu et al. “DrS: Learning Reusable Dense Rewards for Multi-Stage Tasks.” ICLR 2024.\
> > [2] Kim et al. “Subtask-Aware Visual Reward Learning from Segmented Demonstrations.” ICLR 2025.\
> > [3] Belkhale et al. “Data Quality in Imitation Learning.” NeurIPS 2023.\
> > [4] Dass et al. “DataMIL: Selecting Data for Robot Imitation Learning with Datamodels.” arXiv:2505.09603, 2025.

---

> ### Author Response · Authors · 2025-11-27
>
> Dear Reviewer WSRa,
>
> Thank you again for taking the time to review our paper and for engaging with our rebuttal. As we are approaching the end of the discussion phase (12/03), we would greatly appreciate any additional feedback on our response, especially if there are remaining points we should clarify or improve in the next revision.
>
> If our rebuttal has addressed your concerns and you do not have further objections, we would be grateful if you could consider updating your score accordingly. Your feedback is very valuable to us, and we sincerely appreciate your time and effort during the review process.
>
> Sincerely,
>
> The Authors

---

### Official Review · Reviewer_3zBM · 2025-10-31

**Soundness:** 4
**Presentation:** 4
**Contribution:** 3
**Rating:** 4
**Confidence:** 5

**Summary:**

This paper introduces SARM, which introduces stage information about a task to improve reward predictions given a real world robot trajectory/video. The results show this outperforms prior work like ReWiND and aligns closer to the ground truth progress rewards. The learned reward model in SARM further is used to reweight training samples for better performance.

**Strengths:**

- Introduces a novel use of stage prediction to get improved auto generated reward labels of a dataset. This contrasts with prior work that simply used frame indices of a video as the label of the reward progress, which is limited since different stages of the task may proceed at different speeds.
- This work further introduces some useful tricks that improve performance and leads to easier to learn reward models that are much more smooth.
- Strong improvement upon prior work is shown with ablations indicating how the different proposed ideas improve SARM.

**Weaknesses:**

- Compared to prior work this method requires more annotation of datasets (subtask labels).
- There doesn't seem to be an ablation on the reward-model based weighting of training samples, which I believe would be fairly important given the amount of space allocated to describe the RA-BC method. The discussion in 4.3 suggests that a bad reward model would cripple RA-BC results since it may lead to poor reweighting, it may be possible that not doing any re-weighting may end up working better.

**Questions:**

See above

Generally think this work is solid and should be accepted. Happy to raise score if the question about the reward reweighting is addressed.

---

> ### Author Response · Authors · 2025-11-20
> **Response to Reviewer 3zBM**
>
> Dear Reviewer 3zBM,
>
>
> Thank you for your thoughtful comments and for highlighting two important aspects of our work. We address your concerns below.
>
>
> ## (W1) Annotation requirement
>
>
> We thank the reviewer for raising the question regarding annotation cost. SARM is specifically designed to use lightweight subtask boundaries rather than dense, frame-wise progress labels. As shown in Fig. 5–6 and Table A.3, each trajectory only requires marking 3 to 8 subtask transitions, depending on the protocol. This annotation process typically takes 20–40 seconds per demonstration, which is modest relative to dense annotation. Importantly, this one-time lightweight annotation enables two benefits that prior work cannot achieve:
> Resolving large temporal misalignment across demonstrations (Sec. 3.1), which cannot be handled by frame-index heuristics.
> Allowing a single reward model to generalize across heterogeneous trajectory lengths and motion styles, which is essential for long-horizon contact-rich tasks.
>
>
> We also note that subtask-level or language-level annotations are becoming increasingly common in modern open-sourced robotics datasets such as EPIC-Kitchens [1] and Galaxea [2]. Given these trends and the significant improvements in reward stability, OOD generalization, and downstream policy performance, we believe this modest annotation effort is well justified.
>
>
> ## (W2) Ablation on reward-model–based reweighting
>
>
> We appreciate the reviewer’s question and agree that clarifying the effect of reward-model–based sample weighting is important. In the revised manuscript, we updated Sec. 4.2 to explicitly emphasize that Column (1) of Table 3 (also a quick view below) already corresponds to the “no reweighting” ablation the reviewer requested. This provides a direct measurement of how pure BC performs when reweighting is entirely removed. As shown in Table 3, removing reweighting leads to significantly worse performance than RA-BC (whether using ReWiND or SARM), especially on the medium and hard tasks. This clearly demonstrates that the reweighting mechanism is both necessary and effective, and that RA-BC is an essential component for learning from diverse demonstrations.
>
>
> | Training Steps | Tasks  | (1) D_all | (2) D_2min | (3) ReWiND | (4) SARM |
> |---------------:|:------:|:---------:|:----------:|:----------:|:--------:|
> | 20K            | Simple | 12/12     | 12/12      | 12/12      | 12/12    |
> |                | Medium | 0/12      | 4/12       | 1/12       | 7/12     |
> |                | Hard   | 0/12      | 1/12       | 1/12       | 6/12     |
> | 40K            | Simple | 12/12     | 12/12      | 12/12      | 12/12    |
> |                | Medium | 1/12      | 7/12       | 6/12       | 10/12    |
> |                | Hard   | 0/12      | 0/12       | 3/12       | 8/12     |
>
>
> Thank you again for your valuable feedback. We hope these clarifications address your concerns and further strengthen the contribution of our work.
>
>
> Sincerely,\
> The Authors
>
> ---
>
> References:\
> [1] Damen, Dima, et al. “Rescaling Egocentric Vision: Collection, Pipeline and Challenges for EPIC-Kitchens-100.” International Journal of Computer Vision, 130.1 (2022): 33–55.\
> [2] Jiang, Tao, et al. “Galaxea Open-World Dataset and G0 Dual-System VLA Model.” arXiv preprint arXiv:2509.00576 (2025).\

---

> > ### Comment · Reviewer_3zBM · 2025-11-22
> > **Response**
> >
> > Thank you for addressing my concerns. I will raise my score to an 8.
> >
> > I also generally disagree with the other reviewer regarding similarity to past work. This paper is fundamentally quite different to DrS (should still be cited at least) in terms of approach and has real world results as well.

---

> > > ### Author Response · Authors · 2025-11-25
> > >
> > > Thank you very much for the thoughtful follow-up and for raising your score. We greatly appreciate your recognition that our approach is fundamentally different from DrS and that the real-world results strengthen the contribution.
> > >
> > > Thank you again for your constructive feedback throughout the review process.

---

### Official Review · Reviewer_fc5R · 2025-10-31

**Soundness:** 3
**Presentation:** 3
**Contribution:** 3
**Rating:** 8
**Confidence:** 3

**Summary:**

A core challenge in imitation learning literature (e.g. behavior cloning) is learning from diverse demonstration data: multi-modal action distributions, varying teleoperation strategies and proficiency, human errors, as well as sometimes wildly different trajectory lengths depending on the teleoperator and initial/goal configurations. This paper aims to automatically estimate task progress in demonstration data regardless of factors such as trajectory length (recent method ReWiND is prone to fail in this setting), and then subsequently using estimated task progress (reward) to weigh samples during BC policy learning. To do so, the proposed method estimates stage and subtask progress from image observations + joint states + language annotations using a transformer backbone, as well as a number of engineering decisions and heuristics detailed in the paper. Experiments are conducted on a real-world T-shirt folding task using a dual-arm setup, and results indicate that the proposed framework (SARM / RA-BC) improves task progress estimation and downstream policy performance compared to strong baselines.

**Strengths:**

My initial assessment of this paper leans positive overall; it is generally well written, timely, and technically sound. Specifically:
- I believe that this paper studies a relevant and timely problem (learning from "diverse" demonstrations wrt. manipulation strategy, trajectory length etc.), and is likely to be of interest to the community. The problem is clearly motivated in the introduction, and shortcomings of existing work (ReWiND in particular) is described in the related work section. The paper is generally well written and easy to follow, and the illustrations (Figure 1, 2) are helpful for understanding the technical contributions.
- The proposed method seems technically sound and appears to rely fairly little on engineering or manual labor which is definitely a plus (besides language annotation of the demonstration data, I didn't see much discussion on this in the paper but I imagine it may be rather time-consuming). The approach is rather simple and intuitive (I consider this a strength), which makes me believe that it could feasibly be transferred to other tasks and potentially entirely different task domains provided that they can be separated into stages / subtasks.
- I appreciate the focus on a difficult real-world task such as T-shirt folding, and the empirical results on this task are rather compelling. In particular, the qualitative comparison with ReWiND in Figure 3 clearly demonstrates the shortcoming of ReWiND: it is not very robust to varying trajectory lengths and, as a result, struggle with predictions at either end of the completion spectrum. The paper also includes a limited set of results on a dish unloading task (appendix A.6) as well as RL in simulation (pick cube) and the conclusions seem to be rather similar across these tasks and problem settings.

**Weaknesses:**

I do not have any major concerns with the paper in its current form. However, I do believe that there is room for improvement in several ways:
- The writing is a bit odd, with large parts of the experiment section moved to the appendices entirely. It would be helpful if the authors can be more clear about the experimental setup (T-shirt folding, dish unloading, RL finetuning in simulation) and perhaps summarize the results across these different settings in the main paper rather than only discussing them in the appendices; it is easy for readers to miss otherwise.
- I am a little concerned about this particular engineering choice: *"During annotation, only the subtasks defined by the protocol were labeled, and any trajectory that did not contain the complete sequence of subtasks specified by the protocol was discarded. Annotators watched the top-view video and segmented each trajectory into subtasks by recording the start and end frame indices. If a mistake occurred during execution, its start and end frames were also labeled; trajectories containing mistakes were excluded from subsequent model training"* (L184). If one of the main claims of the paper is that SARM / RA-BC can learn from diverse demonstration data, why exclude trajectories that *e.g.* contain a human error somewhere in the trajectory? If the approach was truly robust, it should be able to infer that such actions are indeed errors that then subsequently are corrected by the operator. It would be helpful if the authors can clarify this particular aspect of their approach, and potentially include some experimental results in which data is not filtered as aggressively before any learning; that should give readers a better indication on what the key steps in the data pipeline are.

Minor comments that need to be addressed but are insignificant wrt my overall assessment of the work:
- The abstract is quite long which feels unnecessary, I would suggest revising it to be more concise.
- It would be helpful if the authors could introduce the considered tasks somewhere in the main paper along with e.g. illustrations of what those tasks look like.
- There are a few typos and grammatical errors throughout the paper. While they do not affect understanding, it would be a good idea to proof-read the paper before publication.

**Questions:**

I would really appreciate it if the authors can address my comments in the "weaknesses" section above using written arguments and potentially additional experimental results. My main concerns / questions pertain to writing and the data filtering process.

---

> ### Author Response · Authors · 2025-11-20
> **Response to Reviewer fc5R**
>
> Dear Reviewer fc5R,
>
>
> Thank you for your constructive and detailed feedback. We sincerely appreciate your suggestions and provide clarifications and revisions below.
>
>
> ## (W1, W4) Writing and arrangement
>
>
> We thank the reviewer for the valuable suggestions on writing clarity and section organization. In the revised manuscript, we have moved the dish-unloading experiment—including its setup, results, and analysis—into the main text to ensure that readers do not miss this key evaluation. Additionally, we added a concise introduction to our RL finetuning experiment in the conclusion section, together with a direct pointer to the appendix for experimental details, to make this component more visible and easier to follow.
>
>
> ## (W2) Stage annotation design
>
>
> We thank the reviewer for raising this important question regarding trajectory filtering. We clarify that only trajectories with severe issues—such as the manipulator hitting the table heavily or executing a completely reversed motion sequence—were labeled as “mistakes’’ and removed. These account for less than 2% of the entire dataset. We have revised the paper to clearly describe this criterion. In this paper, “dataset diversity’’ mainly refers to suboptimal but still meaningful trajectories, such as slow, noisy, or unsmooth motions, which we fully retain. Prior works on data curation for imitation learning [1,2] have shown that severely incorrect trajectories can negatively impact BC models. Therefore, for clearly negative-sample trajectories that do not correspond to valid behavior, we believe discarding them is consistent with the underlying logic of imitation learning.
>
>
> ## (W3) Revise abstract
>
>
> We thank the reviewer for the helpful suggestion regarding abstract length. We have rewritten the abstract and reduced it by approximately one third, while ensuring that all essential concepts and findings remain clearly conveyed.
>
>
> ## (W5) Proofreading
>
>
> We thank the reviewer for pointing out the typos and grammatical issues. We have carefully reviewed the full manuscript and corrected multiple issues. A few representative corrections are listed below:\
> (Sec. 1, L093)\
> from: “…a challenging long-horizon task that requires manipulation a deformable objects, where it…”\
> to: “…a challenging long-horizon task that requires manipulating deformable objects, where it…”\
> (Sec. 3.1, L201)\
> from: “Let trajectory i have total length…”\
> to: “Let a trajectory i have total length…”\
> (Sec. 4.2, L427)\
> from: “…the embodiment gap and the lack of high-quality datasets still make it difficult to reproduce…”\
> to: “…the embodiment gap and the lack of high-quality datasets still makes it difficult to reproduce…”\
>
>
> Thank you again for your helpful comments. We hope these clarifications and revisions address your concerns and improve the clarity and quality of the paper.
>
>
> Sincerely,\
> The Authors

---

> > ### Comment · Reviewer_fc5R · 2025-11-24
> > **Thank you**
> >
> > Thank you for responding to my comments in detail. I already leaned towards acceptance (score 8) before the rebuttal and I believe that my main concerns regarding writing and dataset filtering have been addressed in the revised paper so I choose to maintain my score. However, I still believe that the authors should include a few images of their tasks (in particular the dish unloading task) in the main paper to make it more self-contained, this was not addressed in the revision.

---

> > > ### Author Response · Authors · 2025-11-25
> > >
> > > Thank you very much for the clarification and for maintaining your positive assessment. We appreciate your comments on the writing and dataset filtering, and we are glad the revisions addressed your main concerns.
> > >
> > > Regarding the suggestion to include task images—especially for the dish unloading task—we agree that such visuals would make the paper more self-contained. We will consider adding a concise set of task images in a future revision if space allows.
> > >
> > > Thank you again for the constructive feedback and for your support of the paper.

---

### Official Review · Reviewer_pgZ9 · 2025-11-02

**Soundness:** 3
**Presentation:** 4
**Contribution:** 4
**Rating:** 8
**Confidence:** 4

**Summary:**

In this work, the authors introduce a stage-aware, video-based reward modeling framework that jointly predicts high-level task stages and fine-grained progress within each stage. This design effectively addresses the limitations of frame-index-based labeling, which often fails in long, variable-duration tasks such as T-shirt folding. The learned reward models are further utilized to filter high-quality data and reweight training samples based on the estimated rewards. Extensive experiments demonstrate that the proposed reward model outperforms existing baselines in out-of-distribution real-robot rollouts and human demonstration evaluations.

**Strengths:**

1. The authors investigate a very important and timely direction, i.e., how to acquire high-quality data for training imitation learners on long-horizon, contact-rich manipulation tasks such as T-shirt folding. Quantifying demonstration quality is challenging because it depends on latent factors such as action consistency and contact stability, which cannot be directly measured. In this work, the authors propose a video-based reward modeling framework that leverages natural language annotations to assign progress labels and enable stable reward estimation, serving as a mechanism to filter high-quality data and improve policy performance in both simulation and real-world settings. The proposed method is timely and presents a novel idea for learning from suboptimal demonstrations, making it a valuable contribution to the community.

2. The authors provide a comprehensive analysis of the proposed SARM reward model, comparing it against existing methods. The proposed approach demonstrates superior performance across multiple benchmarks. Furthermore, the authors evaluate the framework on  dish unloading, which further validates its effectiveness and generalization capability.

3. By leveraging SARM, the RA-BC policy surpasses both BC baselines by a significant margin on medium and hard tasks. The experiments show that RA-BC effectively exploits diverse datasets by filtering out high-quality data frames, enabling the policy to learn robust, long-horizon manipulation strategies.

4. The authors empirically demonstrate that RA-BC with SARM achieves substantially higher success rates than RA-BC with ReWiND, achieving 83% vs. 50% on medium tasks and 67% vs. 25% on hard tasks, clearly highlighting the advantage of the proposed reward model.

**Weaknesses:**

**Overall Assessment:**

The paper provides a well-justified motivation for the proposed reward model and demonstrates its utility in selecting high-quality data for policy learning. However, several aspects remain unclear and merit further clarification.

1. **Applicability to existing datasets:**

The proposed reward model has shown promise in filtering high-quality data, but it remains unclear how well it generalizes to existing cloth manipulation datasets, such as **DROID* [1] and **Flat’n’Fold** [2]. How would the model perform if applied to these datasets for filtering or reweighting demonstrations?

[1] DROID: A Large-Scale In-the-Wild Robot Manipulation Dataset, 2024
[2] Flat’n’Fold: A Diverse Multi-Modal Dataset for Garment Perception and Manipulation, 2024

2. **Robustness to visual variations:**

How does the proposed reward model handle changes in lighting and illumination conditions? Since real-world environments are often visually diverse and dynamic, it would be valuable to understand the model’s robustness and reliability under such variations.

3. **Baseline comparison:**

The recent work by **Hung et al. (ICLR 2025)** [3] introduces an alternative strategy for learning reward models in long-horizon manipulation tasks. Although the learning objectives differ from those of SARM, a direct comparison between the two approaches under similar long-horizon, contact-rich settings would provide stronger empirical grounding and highlight the unique advantages of the proposed framework.

[3] Hung et al., VICtoR: Learning Hierarchical Vision–Instruction Correlation Rewards for Long-Horizon Manipulation, ICLR 2025

**Questions:**

1. Generalization: How well does the proposed reward model generalize when applied to large-scale, real-world cloth manipulation datasets such as DROID and Flat’n’Fold?

2. Robustness: How robust is the model to lighting and illumination changes, which commonly occur in real-world robotic settings?

3. Comparative Evaluation: How does the proposed method compare with VICtoR (ICLR 2025) in terms of reward modeling effectiveness and policy performance on long-horizon, contact-rich manipulation tasks?

---

> ### Author Response · Authors · 2025-11-20
> **Response to Reviewer pgZ9**
>
> Dear Reviewer pgZ9,
>
>
> Thank you for your detailed and constructive feedback. We sincerely appreciate the opportunity to clarify our contributions and address your concerns. Below we respond to each of your questions point by point.
>
>
> ## (W1Q1) Applicability to existing datasets
>
>
> We appreciate the reviewer’s question regarding the generalization of SARM to existing cloth-manipulation datasets such as DROID [1] and Flat’n’Fold [2]. To the best of our knowledge, DROID does not contain a T-shirt folding task, nor other deformable-object folding tasks that match the long-horizon, multi-stage structure required for SARM’s progress modeling. As a result, it is not suitable for evaluating reward-model-based filtering or progress estimation in our setting.
>
>
> Regarding Flat’n’Fold, although it is a cloth-manipulation dataset, there exists a large mismatch in format, robot embodiment, camera viewpoints, and annotation style. These differences prevent deploying our pretrained SARM model in a zero-shot manner. Furthermore, Flat’n’Fold is relatively new, and to the best of our knowledge, there is currently no published robot behavior model that successfully performs T-shirt folding when trained on it, making direct comparison difficult. We agree that robust cross-dataset reward model transfer is an exciting future direction and plan to explore it in future work.
>
>
> ## (W2Q2) Robustness to visual variations
>
>
> We thank the reviewer for highlighting the importance of robustness to lighting changes. To evaluate SARM under visual perturbations, we added an experiment where each frame’s brightness is modified by up to ±0.3 (noted as “SARM (VB)”). Results are presented in Table 1 as well as the below:
>
>
> | Metrics        | SARM (VB) | SARM  |
> |----------------|-----------|-------|
> | Demo Loss        | 0.015     | 0.009 |
> | Rollout Score    | 0.78      | 0.94  |
> | SE             | 12/12     | 12/12 |
> | PSE            | 10/12     | 11/12 |
> | FE             | 10/12     | 12/12 |
>
>
> SARM exhibits strong robustness under these conditions:
> demonstration loss changes moderately from 0.009 -> 0.015,
> rollout grade decreases from 0.94 -> 0.78, while still outperforming all baselines.
> These results indicate that SARM’s stage-aware temporal modeling provides stable and reliable reward signals even under substantial illumination variation.
>
>
> ## (W3Q3) Baseline comparison with VICtoR (ICLR 2025)
>
>
> We thank the reviewer for pointing out this important baseline. We have now included VICtoR [3] in both the related work discussion and the baseline comparison (Table 1). VICtoR predicts motion class and progress by computing similarity between vision and text embeddings.
>
>
> | Metrics          | VICtoR | SARM  |
> |------------------|--------|-------|
> | Demo Loss          | 0.00   | 0.009 |
> | Rollout Score       | 0.20   | 0.94  |
> | SE               | 0/12   | 12/12 |
> | PSE              | 3/12   | 11/12 |
> | FE               | 7/12   | 12/12 |
>
>
> In our experiments, it produces overly pessimistic reward estimates for the long-horizon T-shirt folding task (rewards rarely exceed 0.5). This suggests that relying solely on pretrained vision–language similarity—without explicitly modeling multi-frame temporal cues—yields reward signals too weak to capture progress in long-horizon, contact-rich manipulation. SARM’s stage-aware decomposition and temporal modeling generate more consistent and fine-grained progress estimates, which explains its stronger empirical performance.
>
>
> Thank you again for your thoughtful comments. We hope our clarifications address your concerns and further demonstrate the strengths and applicability of SARM.
>
>
> Sincerely, \
> The Authors
>
> ---
>
> Reference:\
> [1] Khazatsky, Alexander, et al. "Droid: A large-scale in-the-wild robot manipulation dataset." arXiv preprint arXiv:2403.12945 (2024).\
> [2] Zhuang, Lipeng, et al. "Flat'n'Fold: A Diverse Multi-Modal Dataset for Garment Perception and Manipulation." 2025 IEEE International Conference on Robotics and Automation (ICRA). IEEE, 2025.\
> [3] Hung et al., VICtoR: Learning Hierarchical Vision–Instruction Correlation Rewards for Long-Horizon Manipulation, ICLR 2025

---

> ### Author Response · Authors · 2025-11-27
>
> Dear Reviewer pgZ9,
>
> Thank you again for taking the time to review our paper and for engaging with our rebuttal. As we are approaching the end of the discussion phase (12/03), we would greatly appreciate any additional feedback on our response, especially if there are remaining points we should clarify or improve in the next revision.
>
> Your feedback is very valuable to us, and we sincerely appreciate your time and effort during the review process.
>
> Sincerely,\
> The authors

---

### Author Response · Authors · 2025-11-20
**General response**

Dear Area Chair and Reviewers,


We sincerely thank the AC for handling our paper and all reviewers for their thoughtful and detailed comments. We greatly appreciate the time and insights that each reviewer has provided.


Our paper proposes SARM, a stage-aware reward modeling framework that converts language-annotated demonstrations into smooth, frame-wise progress labels, enabling reliable reward estimation in long-horizon, deformable-object manipulation. Reviewers pgZ9, fc5R, and 3zBM have recognized the contribution as novel, timely, and well evaluated, and all reviewers agree that SARM significantly improves data filtering and downstream policy learning compared to pure BC.


Below, we summarize the key concerns raised by Reviewer WSRa, describe the revisions we have made, and direct reviewers to our detailed point-by-point responses.

(1) Reviewer WSRa raised concerns about the novelty of stage-aware reward modeling in light of DrS [1] and REDS [2].
We have cited both works in the updated manuscript. We have addressed this in the revision and in the detailed responses. In summary:
DrS [1] is a state-based reward model that relies on full simulator states as stage indicators and requires training a separate discriminator for each stage, making it infeasible to apply in our video-based, real-robot setting. Therefore, we have not added DrS experiments, because a direct comparison is difficult and offers limited insight: DrS relies on simulator-state supervision that is unavailable in our real trajectories, and its multi-discriminator design does not scale to our long-horizon visual setting.
REDS [2] learns a semi-sparse step-shaped reward with monotonicity regularization and infers stage from image–subtask embedding similarity. In contrast, SARM explicitly models stage structure and directly predicts a dense, smooth, per-frame progress curve.
We have added a full experimental comparison to REDS, now included in Table 1. We believe these differences in architecture, supervision, and modeling goals constitute a substantial and practically important advancement relative to prior stage-aware reward models. Please see our response to WSRa for the full explanation.

(2) Reviewer WSRa noted that recent work proposes more sophisticated data-quality measures beyond duration-based heuristics. We have revised Lines 51–53 to acknowledge recent advances in data-quality assessment. We now clearly state that, although more sophisticated modeling approaches exist, practical assessment remains challenging in imitation learning. In particular, Belkhale et al. [3] defines data quality via distribution shift between learned-policy and expert-policy state distributions—requiring access to a strong expert and the ability to train/evaluate policies for each proxy. This differs from simple per-trajectory heuristics and is not directly applicable when high-quality demonstrations are scarce. We have added these clarifications and citations in the revised manuscript. Please refer to our detailed response to WSRa for additional discussion.


Across the reviewer responses, we have added or clarified the following items to strengthen the paper:\
(a) Added REDS baseline and analysis (requested by Reviewer WSRa)\
Full comparison to REDS is now included in Table 1. We have analyzed why REDS produces step-shaped, semi-sparse signals that limit performance.\
(b) Added VICtoR baseline (requested by Reviewer pgZ9)\
VICtoR has been added to the experiments (Table 1). We show that embedding-similarity-based progress estimation fails on long-horizon manipulation.\
(c) Robustness to lighting / brightness perturbation (requested by Reviewer pgZ9)\
We have added a stress test with ±0.3 brightness perturbation, showing SARM retains strong performance.


We respectfully invite the AC and Reviewers to refer to the complete point-by-point rebuttals, which provide detailed explanations and all added experimental results.


Thank you again for your valuable feedback. We hope that our revisions, clarifications, and additional experiments address all concerns and further strengthen the paper.


Best regards,\
The Authors

---
References\
[1] Mu et al., DrS: Learning Reusable Dense Rewards for Multi-Stage Tasks, ICLR 2024.\
[2] Kim et al., REDS: Subtask-Aware Visual Reward Learning from Segmented Demonstrations, ICLR 2025.\
[3] Belkhale et al., Data Quality in Imitation Learning, NeurIPS 2023.\
[4] Dass et al., DataMIL: Selecting Data for Robot Imitation Learning with Datamodels, 2025.\
[5] Hung et al., VICtoR: Learning Hierarchical Vision–Instruction Correlation Rewards for Long-Horizon Manipulation, ICLR 2025.

---

### Meta-Review · Area_Chair_dCLe · 2026-01-11

**Summary:**

The paper proposes SARM, a stage-aware reward modeling framework for long-horizon robotic manipulation that jointly predicts task stage and fine-grained progress using natural-language subtask annotations.

The key concerns raised across reviews were:

Novelty and positioning (WSRa): Whether stage-aware reward modeling represents sufficient novelty given prior work such as DrS and REDS, and whether the data-quality literature was adequately addressed.

Baseline comparisons (pgZ9, WSRa): Requests for comparison with VICtoR and REDS to strengthen empirical grounding.

Robustness (pgZ9): Whether the reward model handles visual variations such as lighting changes in real-world settings.

Ablation clarity (3zBM): Whether the contribution of reward-based sample reweighting was properly ablated.

Writing and presentation (fc5R): Organization of experimental sections and clarity of the data filtering process.

**Reviewer Concerns:**

The authors provided a comprehensive rebuttal that addressed all actionable concerns:

REDS and VICtoR baselines: Both were added to Table 1 with detailed analysis. REDS produces step-shaped, semi-sparse signals that underperform SARM (Rollout Score 0.16 vs 0.94). VICtoR's embedding-similarity approach yields overly pessimistic rewards (0.20 vs 0.94), demonstrating limitations of approaches lacking explicit temporal modeling.

Robustness to visual perturbations: A ±0.3 brightness stress test was added, showing SARM maintains strong performance (Rollout Score 0.78 under perturbation vs 0.94 baseline), still outperforming all baselines.

Ablation on reweighting: The authors clarified that Table 3, Column 1 (D_all) already represents the "no reweighting" ablation, showing significant performance degradation (0/12 on Medium/Hard tasks vs 10/12 and 8/12 with SARM reweighting).

Data-quality literature: The revision now cites Belkhale et al. (NeurIPS 2023) and Dass et al. (2025), clarifying that these methods require policy training and expert access, unlike simple per-trajectory heuristics.

Writing improvements: The dish-unloading experiment was moved to the main text, the abstract was shortened, and typos were corrected.

Outstanding Concerns

Novelty relative to DrS (WSRa): While the authors explain that DrS requires simulator states and per-stage discriminators (making direct comparison infeasible), this concern remains partially subjective. However, Reviewer 3zBM explicitly disagreed with this criticism, stating the paper is "fundamentally quite different to DrS in terms of approach and has real-world results."

Generalization to external datasets (pgZ9): The authors acknowledged that zero-shot transfer to datasets like DROID or Flat'n'Fold is not feasible due to domain gaps, positioning cross-dataset transfer as future work. This is reasonable given the scope of the current contribution.

**Reviewer Scores:**

Reviewer pgZ9 (Original: 8): Would likely maintain score at 8. All three concerns (generalization, robustness, VICtoR comparison) were addressed with experiments or reasonable explanations. The reviewer did not respond after the rebuttal but had no fundamental objections.

Reviewer fc5R (Original: 8): Explicitly confirmed score of 8 post-rebuttal, stating main concerns on writing and dataset filtering were addressed.

Reviewer 3zBM (Original: 4  Updated: 8): Already raised score to 8 during discussion after ablation clarification. Explicitly supported acceptance and disagreed with WSRa's novelty concerns.

Reviewer WSRa (Original: 2): Did not participate in discussion. Given the thorough rebuttal—including REDS experiments, literature citations, and architectural clarifications—a reasonable increase to 4-5 might be expected, though the reviewer's silence makes this uncertain. The core novelty concern is contested by another expert reviewer (3zBM).

The paper makes a solid contribution to reward modeling for long-horizon manipulation with strong real-world results (83% success on T-shirt folding vs 8% for vanilla BC). Three of four reviewers support acceptance with scores of 8, and the lone dissenting reviewer did not engage with the substantive rebuttal. The authors' additions of REDS and VICtoR comparisons, robustness experiments, and literature clarifications strengthen the empirical foundation. The novelty concern is addressed through clear differentiation from prior work (video-based vs state-based, dense vs semi-sparse rewards) and is contested by an expert reviewer familiar with this literature. The paper represents a timely, well-executed contribution to robotic manipulation that will be of interest to the ICLR community.

---

### Decision · Program_Chairs · 2026-01-26

Accept (Poster)